# Adaptive Discretization for Continuous Control using Particle Filtering Policy Network

## Abstract

Controlling the movements of highly articulated agents and robots has been a long-standing challenge to model-free deep reinforcement learning. In this paper, we propose a simple, yet general, framework for improving the performance of policy gradient algorithms by discretizing the continuous action space. Instead of using a fixed set of predetermined atomic actions, we exploit particle filtering to adaptively discretize actions during training and track the posterior policy represented as a mixture distribution. The resulting policy can replace the original continuous policy of any given policy gradient algorithm without changing its underlying model architecture. We demonstrate the applicability of our approach to state-of-the-art on-policy and off-policy baselines in challenging control tasks. Baselines using our particle-based policies achieve better final performance and speed of convergence as compared to corresponding continuous implementations and implementations that rely on fixed discretization schemes.

## 1 Introduction

In the last few years, impressive results have been obtained by deep reinforcement learning (DRL) both on physical and simulated articulated agents for a wide range of motor tasks that involve learning controls in high-dimensional continuous action spaces (Lillicrap et al., 2015; Levine et al., 2016; Heess et al., 2017; Haarnoja et al., 2018c; Rajeswaran et al., 2018; Tan et al., 2018; Peng et al., 2018; 2020). Many methods have been proposed that can improve the performance of DRL for continuous control problems, e.g. distributed training (Mnih et al., 2016; Espeholt et al., 2018), hierarchical learning (Daniel et al., 2012; Haarnoja et al., 2018a), and maximum entropy regularization (Haarnoja et al., 2017; Liu et al., 2017; Haarnoja et al., 2018b). Most of such works, though, focus on learning mechanisms to boost performance beyond the basic distribution that defines the action policy, where a Gaussian-based policy or that with a squashing function is the most common choice as the basic policy to deal with continuous action spaces. However, the unimodal form of Gaussian distributions could experience difficulties when facing a multi-modal reward landscape during optimization and prematurely commit to suboptimal actions (Daniel et al., 2012; Haarnoja et al., 2017).

To address the unimodality issue of Gaussian policies, people have been exploring more expressive distributions than Gaussians, with a simple solution being to discretize the action space and use categorical distributions as multi-modal action policies (Andrychowicz et al., 2020; Jaśkowski et al., 2018; Tang & Agrawal, 2019). However, categorical distributions cannot be directly extended to many off-policy frameworks as their sampling process is not reparameterizable. Importantly, the performance of the action space discretization depends a lot on the choice of discrete atomic actions, which are usually picked uniformly due to lack of prior knowledge. On the surface, increasing the resolution of the discretized action space can make fine control more possible. However, in practice, this can be detrimental to the optimization during training, since the policy gradient variance increases with increasing number of atomic actions (Tang & Agrawal, 2019).

Our work also focuses on action policies defined by an expressive, multimodal distribution. Instead of selecting fixed samples from the continuous action space, though, we exploit a particle-based approach to sample the action space dynamically during training and track the policy represented as a mixture distribution with state-independent components. We refer to the resulting policy network as Particle Filtering Policy Network (PFPN). We evaluate PFPN on state-of-the-art on-policy and off-policy baselines using high-dimensional tasks from the PyBullet Roboschool en-

vironments (Coumans & Bai, 2016–2019) and the more challenging DeepMimic framework (Peng et al., 2018). Our experiments show that baselines using PFPN exhibit better overall performance and/or speed of convergence and lead to more robust agent control. as compared to uniform discretization and to corresponding implementations with Gaussian policies.

**Main Contributions.** Overall, we make the following contributions. We propose PFPN as a general framework for providing expressive action policies dealing with continuous action spaces. PFPN uses state-independent particles to represent atomic actions and optimizes their placement to meet the fine control demand of continuous control problems. We introduce a reparameterization trick that allows PFPN to be applicable to both on-policy and off-policy policy gradient methods. PFPN outperforms unimodal Gaussian policies and the uniform discretization scheme, and is more sample-efficient and stable across different training trials. In addition, it leads to high quality motion and generates controls that are more robust to external perturbations. Our work does not change the underlying model architecture or learning mechanisms of policy gradient algorithms and thus can be applied to most commonly used policy gradient algorithms.

## 2 BACKGROUND

We consider a standard reinforcement learning setup where given a time horizon $H$ and the trajectory $\tau = (\mathbf{s}_1, \mathbf{a}_1, \cdots, \mathbf{s}_H, \mathbf{a}_H)$ obtained by a transient model $\mathcal{M}(\mathbf{s}_{t+1}|\mathbf{s}_t, \mathbf{a}_t)$ and a parameterized action policy $\pi_\theta(\mathbf{a}_t|\mathbf{s}_t)$, with $\mathbf{s}_t \in \mathbb{R}^n$ and $\mathbf{a}_t \in \mathbb{R}^m$ denoting the state and action taken at time step $t$, respectively, the goal of learning is to optimize $\theta$ that maximize the cumulative reward:

$$J(\theta) = \mathbb{E}_{\tau \sim p_\theta(\tau)}\left[r_t(\tau)\right] = \int p_\theta(\tau) r(\tau) d\tau. \tag{1}$$

Here, $p_\theta(\tau)$ denotes the state-action visitation distribution for the trajectory $\tau$ induced by the transient model $\mathcal{M}$ and the action policy $\pi_\theta$ with parameter $\theta$, and $r(\tau) = \sum_t r(\mathbf{s}_t, \mathbf{a}_t)$ where $r(\mathbf{s}_t, \mathbf{a}_t) \in \mathbb{R}$ is the reward received at time step $t$. We can maximize $J(\theta)$ by adjusting the policy parameters $\theta$ through the gradient ascent method, where the gradient of the expected reward can be determined according to the policy gradient theorem (Sutton et al., 2000), i.e.

$$\nabla_\theta J(\theta) = \mathbb{E}_{\tau \sim \pi_\theta(\cdot|\mathbf{s}_t)}\left[A_t \nabla_\theta \log \pi_\theta(\mathbf{a}_t|\mathbf{s}_t)|\mathbf{s}_t\right]. \tag{2}$$

where $A_t \in \mathbb{R}$ denotes an estimate to the reward term $r_t(\tau)$. In DRL, the estimator of $A_t$ often relies on a separate network (critic) that is updated in tandem with the policy network (actor). This gives rise to a family of policy gradient algorithms known as actor-critic.

**On-Policy and Off-Policy Actor-Critics.** In on-policy learning, the update policy is also the behavior policy based on which a trajectory is obtained to estimate $A_t$. Common on-policy actor-critic algorithms include A3C (Mnih et al., 2016) and PPO (Schulman et al., 2017), and directly employ Equation 2 for optimization. In off-policy learning, the policy can be updated without the knowledge of a whole trajectory. This results in more sample efficient approaches as samples are reusable. While algorithms such as Retrace (Munos et al., 2016) and PCL (Nachum et al., 2017) rely on Equation 2, many off-policy algorithms exploit a critic network to estimate $A_t$ given a state-action pair (Q- or soft Q-value). Common off-policy actor-critic methods include DDPG (Lillicrap et al., 2015), SAC (Haarnoja et al., 2018b;d) and their variants (Haarnoja et al., 2017; Fujimoto et al., 2018). These methods perform optimization to maximize a state-action value $Q(\mathbf{s}_t, \mathbf{a}_t)$. In order to update the policy network with parameter $\theta$, they require the action policy to be reparameterizable such that the sampled action $\mathbf{a}_t$ can be rewritten as a function differentiable to the parameter $\theta$, and the optimization can be done through the gradient $\nabla_{\mathbf{a}_t} Q(\mathbf{s}_t, \mathbf{a}_t) \nabla_\theta \mathbf{a}_t$.

**Policy Representation.** Given a multi-dimensional continuous action space, the most common choice in current DRL baselines is to model the policy $\pi_\theta$ as a multivariate Gaussian distribution with independent components for each action dimension (DDPG, SAC and their variants typically use Gaussian with a monotonic squashing function to stabilize the training). For simplicity, let us consider a simple case with a single action dimension and define the action policy as $\pi_\theta(\cdot|\mathbf{s}_t) := \mathcal{N}(\mu_\theta(\mathbf{s}_t), \sigma_\theta^2(\mathbf{s}_t))$. Then, we can obtain $\log \pi_\theta(a_t|\mathbf{s}_t) \propto -(a_t - \mu_\theta(\mathbf{s}_t))^2$. Given a sampled action $a_t$ and the estimate of cumulative rewards $A_t$, the optimization process based on the above expression can be imagined as that of shifting $\mu_\theta(\mathbf{s}_t)$ towards the direction of $a_t$ if $A_t$ is higher

than the expectation, or to the opposite direction if $A_t$ is smaller. Such an approach, though, can easily converge to a suboptimal solution, if, for example, the reward landscape has a basis between the current location of $\mu_\theta(\mathbf{s}_t)$ and the optimal solution, or hard to be optimized if the reward landscape is symmetric around $\mu_\theta(\mathbf{s}_t)$. These issues arise due to the fact that the Gaussian distribution is inherently unimodal, while the reward landscape could be multi-modal (Haarnoja et al., 2017). Similar problems also could occur in Q-value based optimization, like DDPG and SAC. We refer to Appendix F for further discussion about the limitations of unimodal Gaussian policies and the value of expressive multimodal policies.

## 3 PARTICLE FILTERING POLICY NETWORK

In this section, we describe our Particle Filtering Policy Network (PFPN) that addresses the unimodality issues from which typical Gaussian-based policy networks suffer. Our approach represents the action policy as a mixture distribution obtained by adaptively discretizing the action space using state-independent particles, each capturing a Gaussian distribution. The policy network, instead of directly generating actions, it is tasked with choosing particles, while the final actions are obtained by sampling from the selected particles.

### 3.1 PARTICLE-BASED ACTION POLICY

Generally, we define $\mathcal{P} := \{\langle \mu_{i,k}, w_{i,k}(\mathbf{s}_t|\theta)\rangle | i = 1, \cdots, n; k = 1, \cdots, m\}$ as a weighted set of particles for continuous control problems with a $m$-dimension action space and $n$ particles distributed on each action space, where $\mu_{i,k} \in \mathbb{R}$ representing an atomic action location on the $k$-th dimension of the action space, and $w_{i,k}(\mathbf{s}_t|\theta)$, satisfying $\sum_i w_{i,k}(\mathbf{s}_t|\theta) = 1$, denotes the associated weight generated by the policy network with parameter $\theta$ given the input state $\mathbf{s}_t$. Let $p_{i,k}(a_{i,k}|\mu_{i,k}, \xi_{i,k})$ denote the probability density function of the distribution defined by the location $\mu_{i,k}$ and a noise process $\xi_{i,k}$. Given $\mathcal{P}$, we define the action policy as factorized across the action dimensions:

$$\pi_\theta^{\mathcal{P}}(\mathbf{a}_t|\mathbf{s}_t) = \prod_k \sum_i w_{i,k}(\mathbf{s}_t|\theta) p_{i,k}(a_{t,k}|\mu_{i,k}, \xi_{i,k}), \tag{3}$$

where $\mathbf{a}_t = \{a_{t,1}, \cdots, a_{t,m}\}$, $a_{t,k} \in \mathbb{R}$ is the sampled action at the time step $t$ for the action dimension $k$, and $w_{i,k}(\cdot|\theta)$ can be obtained by applying a softmax operation to the output neurons of the policy network for the $k$-th dimension. The state-independent parameter set, $\{\mu_{i,k}\}$, gives us an adaptive discretization scheme that can be optimized during training. The choice of noise $\xi_{i,k}$ relies on certain algorithms. It can be a scalar, e.g., the Ornstein–Uhlenbeck noise in DDPG (Lillicrap et al., 2015) or an independent sample drawn from the standard normal distribution $\mathcal{N}(0, 1)$ in soft Q-learning (Haarnoja et al., 2017), or be decided by a learnable variable, for example, a sample drawn from $\mathcal{N}(0, \xi_{i,k}^2)$ with a learnable standard deviation $\xi_{i,k}$. In the later case, a particle become a Gaussian component $\mathcal{N}(\mu_{i,k}, \xi_{i,k}^2)$. Without loss of generality, we define the parameters of a particle as $\phi_{i,k} = [\mu_{i,k}, \xi_{i,k}]$ for the following discussion.

While the softmax operation gives us a categorical distribution defined by $w_{\cdot,k}(\mathbf{s}_t|\theta)$, the nature of the policy for each dimension is a mixture distribution with state-independent components defined by $\phi_{i,k}$. The number of output neurons in PFPN increases linearly with the increase in the number of action dimensions and thus makes it suitable for high-dimensional control problems. Drawing samples from the mixture distribution can be done in two steps: first, based on the weights $w_{\cdot,k}(\mathbf{s}_t|\theta)$, we perform sampling on the categorical distribution to choose a particle $j_k$ for each dimension $k$, i.e. $j_k(\mathbf{s}_t) \sim P(\cdot|w_{\cdot,k}(\mathbf{s}_t))$; then, we can draw actions from the components represented by the chosen particles with noise as $a_{t,k} \sim p_{j_k(\mathbf{s}_t)}(\cdot|\phi_{j_k(\mathbf{s}_t)})$.

Certain algorithms, like A3C and IMPALA, introduce differential entropy loss to encourage exploration. However, it may be infeasible to analytically evaluate the differential entropy of a mixture distribution without approximation (Huber et al., 2008). As such, we use the entropy of the categorical distribution if a differential entropy term is needed during optimization.

We refer to Appendix C for the action policy representation in DDPG and SAC where an invertible squashing function is applied to Gaussian components.

## 3.2 TRAINING

The proposed particle-based policy distribution is general and can be applied directly to any algorithm using the policy gradient method with Equation 2. To initialize the training, due to lack of prior knowledge, the particles can be distributed uniformly along the action dimensions with a standard deviation covering the gap between two successive particles. With no loss of generality, let us consider below only one action dimension and drop the subscript $k$. Then, at every training step, each particle $i$ will move along its action dimension and be updated by

$$\nabla J(\phi_i) = \mathbb{E}\left[\sum_t c_t w_i(s_t|\theta)\nabla_{\phi_i} p_i(a_t|\phi_i)|\mathbf{s}_t\right] \tag{4}$$

where $a_t \sim \pi_\theta^{\mathcal{P}}(\cdot|\mathbf{s}_t)$ is the action chosen during sampling, and $c_t = \frac{A_t}{\sum_j w_j(\mathbf{s}_t|\theta)p_j(a_t|\phi_j)}$ is a coefficient shared by all particles on the same action dimension. Our approach focuses only on the action policy representation in general policy gradient methods. The estimation of $A_t$ can be chosen as required by the underlying policy gradient method, e.g. the generalized advantage estimator (Schulman et al., 2015b) in PPO/DPPO and the V-trace based temporal difference error in IMPALA. Similarly, for the update of the policy neural network, we have

$$\nabla J(\theta) = \mathbb{E}\left[\sum_t c_t p_i(a_t|\phi_i)\nabla_\theta w_i(\mathbf{s}_t|\theta)|\mathbf{s}_t\right]. \tag{5}$$

From the above equations, although sampling is performed on only one particle for each given dimension, all of that dimension's particles will be updated during each training iteration to move towards or away from the location of $a_t$ according to $A_t$. The amount of the update, however, is regulated by the state-dependent weight $w_i(\mathbf{s}_t|\theta)$: particles that have small probabilities to be chosen for a given state $\mathbf{s}_t$ will be considered as uninteresting and be updated with a smaller step size or not be updated at all. On the other hand, the contribution of weights is limited by the distance between a particle and the sampled action: particles too far away from the sampled action would be considered as insignificant to merit any weight gain or loss. In summary, particles can converge to different optimal locations near them during training and be distributed multimodally according to the reward landscape defined by $A_t$, rather than collapsing to a unimodal, Gaussian-like distribution.

## 3.3 RESAMPLING

Similar to traditional particle filtering approaches, our approach would encounter the problem of degeneracy (Kong et al., 1994). During training, a particle placed near a location at which sampling gives a low $A_t$ value would achieve a weight decrease. Once the associated weight reaches near zero, the particle will not be updated anymore (cf. Equation 4) and become 'dead'. We adapt the idea of importance resampling from the particle filtering literature (Doucet et al., 2001) to perform resampling for dead particles and reactivate them by duplicating alive target particles.

We consider a particle dead if its maximum weight over all possible states is too small, i.e. $\max_{\mathbf{s}_t} w_i(\mathbf{s}_t|\theta) < \epsilon$, where $\epsilon$ is a small positive threshold number. In practice, we cannot check $w_i(s_t|\theta)$ for all possible states, but can keep tracking it during sampling based on the observed states collected in the last batch of environment steps. A target particle $\tau_i$ is drawn for each dead particle $i$ independently from the categorical distribution based on the particle's average weight over the observed samples: $\tau_i \sim P(\cdot|\mathbb{E}_{\mathbf{s}_t}[w_k(\mathbf{s}_t|\theta)], k = 1, 2, \cdots)$.

**Theorem 1.** *Let $\mathcal{D}_\tau$ be a set of dead particles sharing the same target particle $\tau$. Let also the logits for the weight of each particle $k$ be generated by a fully-connected layer with parameters $\boldsymbol{\omega}_k$ for the weight and $b_k$ for the bias. The policy $\pi_\theta^{\mathcal{P}}(a_t|\mathbf{s}_t)$ is guaranteed to remain unchanged after resampling via duplicating $\phi_i \leftarrow \phi_\tau, \forall i \in D_\tau$, if the weight and bias used to generate the unnormalized logits of the target particle are shared with those of the dead one as follows:*

$$\boldsymbol{\omega}_i \leftarrow \boldsymbol{\omega}_\tau; \quad b_i, b_\tau \leftarrow b_\tau - \log(|\mathcal{D}_\tau| + 1). \tag{6}$$

*Proof.* See Appendix B for the inference. □

Theorem 1 guarantees the correctness of our resampling process as it keep the action policy $\pi_\theta^{\mathcal{P}}(a_t|\mathbf{s}_t)$ identical before and after resampling. If, however, two particles are exactly the same

after resampling, they will always be updated together at the same pace during training and lose diversity. To address this issue, we add some regularization noise to the mean value when performing resampling, i.e. $\mu_i \leftarrow \mu_\tau + \varepsilon_i$, where $\varepsilon_i$ is a small random number to prevent $\mu_i$ from being too close to its target $\mu_\tau$.

### 3.4 REPARAMETERIZATION TRICK

The two-step sampling method described in Section 3.1 is non-reparameterizable, because of the standard way of sampling from the categorical distribution through which Gaussians are mixed. To address this issue and enable the proposed action policy applicable in state-action value based off-policy algorithms, we consider the concrete distribution (Jang et al., 2016; Maddison et al., 2016) that generates a reparameterized continuous approximation to a categorical distribution. We refer to Appendix D for a formal definition of concrete distribution defined as CONCRETE below.

Let $\mathbf{x}(\mathbf{s}_t|\theta) \sim \text{CONCRETE}(\{w_i(\mathbf{s}_t|\theta); i = 1, 2, \cdots\}, 1)$, where $\mathbf{x}(\mathbf{s}_t|\theta) = \{x_i(\mathbf{s}_t|\theta); i = 1, 2, \cdots\}$ is a is a reparametrizable sampling result of a relaxed version of the one-hot categorical distribution supported by the probability of $\{w_i(\mathbf{s}_t|\theta); i = 1, 2, \cdots\}$. We apply the Gumbel-softmax trick (Jang et al., 2017) to get a sampled action value as

$$a'(\mathbf{s}_t) = \text{STOP}\left(\sum_i a_i \delta(i, \arg\max \mathbf{x}(\mathbf{s}_t|\theta))\right) \tag{7}$$

where $a_i$ is the sample drawn from the distribution represented by the particle $i$ with parameter $\phi_i$, STOP$(\cdot)$ is a "gradient stop" operation, and $\delta(\cdot, \cdot)$ denotes the Kronecker delta function. Then, the reparameterized sampling result can be written as follows:

$$a_\theta^{\mathcal{P}}(\mathbf{s}_t) = \sum_i (a_i - a'(\mathbf{s}_t))m_i + a'(\mathbf{s}_t)\delta(i, \arg\max \mathbf{x}) \equiv a'(\mathbf{s}_t), \tag{8}$$

where $m_i := x_i(\mathbf{s}_t|\theta) + \text{STOP}(\delta(i, \arg\max \mathbf{x}(\mathbf{s}_t|\theta)) - x_i(\mathbf{s}_t|\theta)) \equiv \delta(i, \arg\max \mathbf{x}(\mathbf{s}_t|\theta))$ composing a one-hot vector that approximates the samples drawn from the corresponding categorical distribution. Since $x_i(\mathbf{s}_t|\theta)$ drawn from the concrete distribution is differentiable to the parameter $\theta$, the gradient of the reparameterized action sample can be obtained by

$$\nabla_\theta a_\theta^{\mathcal{P}}(\mathbf{s}_t) = \sum_i (a_i - a'(\mathbf{s}_t))\nabla_\theta x_i(\mathbf{s}_t|\theta); \nabla_{\phi_i} a_\theta^{\mathcal{P}} = \delta(i, \arg\max \mathbf{x}(\mathbf{s}_t|\theta))\nabla_{\phi_i} a_i. \tag{9}$$

Through these equations, both the policy network parameter $\theta$ and the particle parameters $\phi_i$ can be updated by backpropagation through the sampled action $a'(\mathbf{s}_t)$.

## 4 RELATED WORK

Our approach focuses on the action policy representation exploiting a more expressive distribution other than Gaussians for continuous control problem in DRL using policy gradient method. In on-policy gradient methods, action space discretization using a categorical distribution to replace the original, continuous one has been successfully applied in some control tasks (Andrychowicz et al., 2020; Tang & Agrawal, 2019). All of these works discretize the action space uniformly and convert the action space to a discrete one before training. While impressive results have been obtained that allow for better performance, such a uniform discretization scheme heavily relies on the number of discretized atomic actions to find a good solution. On the other hand, discretized action spaces cannot be directly applied on many off-policy policy gradient methods, since categorical distributions are non-reparameterizable. While DQN-like approaches are efficient for problems with discrete action spaces, such techniques without policy networks cannot scale well to high-dimensional continuous action spaces due to the curse of dimensionality (Lillicrap et al., 2015). Recent work attempted to solve this issue by using a sequential model, at the expense, though, of increasing the complexity of the state space (Metz et al., 2017). Chou et al. (2017) proposed to use beta distributions as a replacement to Gaussians. However, results from Tang & Agrawal (2018a) show that beta distributions do not work well as Gaussians in high-dimensional tasks.

Our PFPN approach adopts a mixture distribution as the action policy, which becomes a mixture of Gaussians when a learnable standard deviation variable is introduced to generate a normal noise

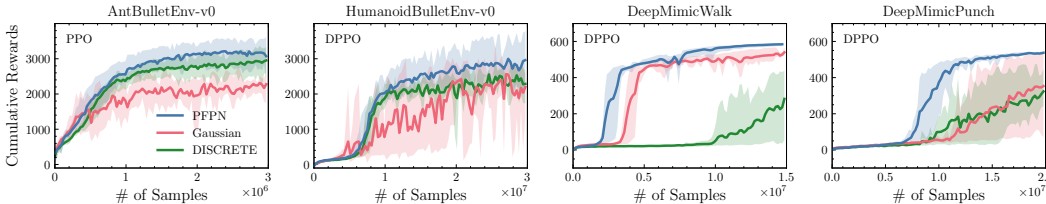

Figure 1: Learning curves of PPO/DPPO using PFPN (blue) compared to Gaussian policies (red) and DISCRETE policies (green). Solid lines report the average and shaded regions are the minimum and maximum cumulative rewards achieved with different random seeds during training.

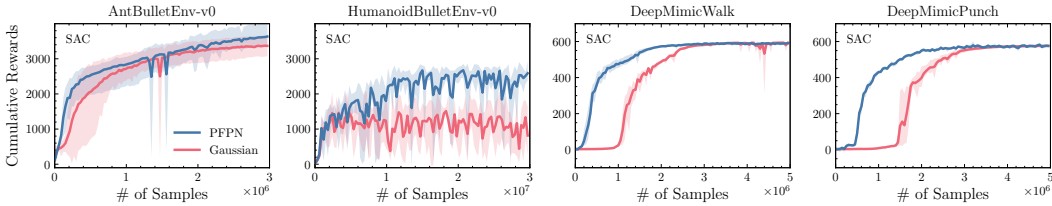

Figure 2: Comparison of SAC using PFPN (blue) and Gaussian policies (red). DISCRETE is not shown as the categorical distribution cannot be applied to SAC.

for sampling. It uses state-independent particles having state-dependent weights to track the policy distribution. While an early version of SAC also employed a mixture of Gaussians, such a mixture is completely state dependent, and hence it cannot provide a discretization scheme that is consistent globally for all given states. In addition, the components in the fully state-dependent mixture of Gaussians could collapse, resulting in similar issues as unimodal Gaussian policies (Tang & Agrawal, 2018a). Adopting state-independent components can reduce the network size and still provide expressive action policies when sufficient components are employed.

PFPN focuses on the policy distribution representation generated directly by the policy network without changing the underlying policy gradient algorithms or remodeling the problem, and thus it can be applied directly to most common used policy gradient methods in DRL. It is also complementary to recent works that focus on improving the expressiveness of the action policy through normalizing flows (Haarnoja et al., 2018a; Tang & Agrawal, 2018b; Mazoure et al., 2019; Delalleau et al., 2019), where the mixture distribution provided by PFPN can be employed as a base distribution. Other techniques applicable to policy gradient methods with a generally defined basic action policy can be combined with PFPN as well, such as the use of ordinal architecture for action parameterization (Tang & Agrawal, 2019), action space momentum as in the recent PPO-CMA (Hämäläinen et al., 2018), and energy-based policy distribution optimization methods, like PGQL (O'Donoghue et al., 2016), Soft-Q learning (Haarnoja et al., 2017), SVPG (Liu et al., 2017), and policy optimization using Wasserstein Gradient Flows (Zhang et al., 2018).

## 5 EXPERIMENTS

The goal of our experiments is to evaluate whether existing policy gradient algorithms using PFPN can outperform the corresponding implementations with Gaussian policies, along with comparing our adaptive action discretization scheme generated by PFPN to the fixed, uniform one. For our comparisons, we use the set of particles as a mixture of Gaussians with learnable standard deviation. We run benchmarks on a range of continuous torque-based control tasks from PyBullet Roboschool environments (Schulman et al., 2015a). We also consider several challenging position-control tasks from the DeepMimic framework (Peng et al., 2018) where a 36-dimension humanoid agent learns a locomotion policy based on motion capture data with a 197-dimension state space.

### 5.1 COMPARISONS

Figure 1 and 2 evaluate our approach on two representative policy gradient methods: PPO/DPPO (Schulman et al., 2017; Heess et al., 2017), which is a stable on-policy method that exhibits good performance, and SAC (Haarnoja et al., 2018d), an off-policy method that achieves state-of-the-art performance in many tasks. The figures show the learning curve by cumulative rewards of evaluation rollouts during training. We also compare PFPN to the fixed discretization scheme (DISCRETE) obtained by uniformly discretizing each action dimension into a fixed number of bins and sampling actions from a categorical distribution. In all comparisons, PFPN and DISCRETE exploit the same number of atomic actions, i.e., number of particles in PFPN and number of bins in DISCRETE. We train five trials of each baseline with different random seeds that are the same across PFPN and the corresponding implementations of other methods. Evaluation was performed ten times every 1,000 training steps using deterministic action.

As it can be seen in Figure 1, PFPN outperforms Gaussian policies and DISCRETE in all of these tested tasks. Compared to Gaussian policies, our particle-based scheme achieves better final performance and typically exhibits faster convergence while being more stable across multiple trials. In the Roboschool tasks, DISCRETE performs better than the Gaussian policies. However, this doesn't translate to the DeepMimic tasks where fine control demand is needed to reach DeepMimic's multimodal reward landscape, with DISCRETE showing high variance and an asymptotic performance that is on par with or worse than Gaussian-based policies. In contrast, PFPN is considerably faster and can reach stable performance that is higher than both Gaussian and DISCRETE policies.

In the SAC benchmarks shown in Figure 2, DISCRETE cannot be applied since the categorical distribution that it employs as the action policy is non-parameterizable. PFPN works by exploiting the reparameterization trick detailed in Section 3.4. Given the state-of-the-art performance of SAC, the PFPN version of SAC performs comparably to or better than the vanilla SAC baseline and has faster convergence in most of those tasks, which demonstrates the effectiveness of our proposed adaptive discretization scheme. In DeepMimic tasks, considering the computation cost of running stable PD controllers to determine the torque applied to each humanoid joint (Tan et al., 2011), PFPN can save hours of training time due to its sampling efficiency.

PFPN can be applied to currently popular policy-gradient DRL algorithms. We refer to Appendix H for additional benchmark results, as well as results obtained with A2C/A3C (Mnih et al., 2016), IMPALA (Espeholt et al., 2018), and DDPG (Lillicrap et al., 2015). In most of these benchmarks, PFPN outperforms the Gaussian baselines and DISCRETE scheme, and is more stable achieving similar performance across different training trials. In Appendix H.5, we also compare the performance in terms of wall clock time, highlighting the sampling efficiency of PFPN over Gaussian and DISCRETE when facing complex control tasks. See Appendix G for all hyperparameters.

### 5.2 ADAPTIVE DISCRETIZATION

| particles/ bins | AntBulletEnv-v0 | | HumanoidBulletEnv-v0 | | DeepMimicWalk | | DeepMimicPunch | |
|---|---|---|---|---|---|---|---|---|
| | PFPN | DISCRETE | PFPN | DISCRETE | PFPN | DISCRETE | PFPN | DISCRETE |
| 5 | **3154 ± 209** | 2958 ± 147 | **2568 ± 293** | 2567 ± 416 | **438 ± 15** | 61 ± 96 | **37 ± 15** | 10 ± 1 |
| 10 | **3163 ± 323** | 2863 ± 281 | **2840 ± 480** | 2351 ± 343 | **489 ± 16** | 308 ± 86 | **426 ± 48** | 281 ± 155 |
| 35 | **2597 ± 246** | 2367 ± 274 | **2276 ± 293** | 2255 ± 376 | **584 ± 4** | 245 ± 164 | **537 ± 7** | 317 ± 76 |
| 50 | **2571 ± 163** | 2310 ± 239 | **2191 ± 322** | 1983 ± 325 | **580 ± 6** | 322 ± 195 | **521 ± 19** | 198 ± 159 |
| 100 | **2234 ± 104** | 2181 ± 175 | **1444 ± 330** | 1427 ± 358 | **579 ± 18** | 277 ± 197 | **533 ± 7** | 224 ± 164 |
| 150 | **2335 ± 147** | 2114 ± 160 | **1164 ± 323** | 1084 ± 453 | **583 ± 13** | 294 ± 200 | **531 ± 17** | 180 ± 152 |
| 200 | - | | - | | **583 ± 15** | 360 ± 166 | **509 ± 31** | 181 ± 153 |
| 400 | - | | - | | **578 ± 14** | 111 ± 159 | **478 ± 63** | 126 ± 137 |
| Gaussian | 2327 ± 199 | | 2462 ± 195 | | 540 ± 19 | | 359 ± 181 | |

Table 1: Comparison between PFPN and DISCRETE on four benchmarks using PPO/DPPO while varying the resolution of each action dimension. Training stops when a fixed number of samples is met as shown in Figure 1. Reported numbers denote final performance averaged over 5 trials ± std.

Compared to Gaussian-based policy networks, DISCRETE can work quite well in many on-policy tasks, as has been shown in recent prior work (Tang & Agrawal, 2019). However, in the comparative evaluations outlined above, we showed that the adaptive discretization scheme that PFPN employs results in higher asymptotic performance and/or faster convergence as compared to the uniform discretization scheme. To gain a better understanding of the advantages of adaptive discretization

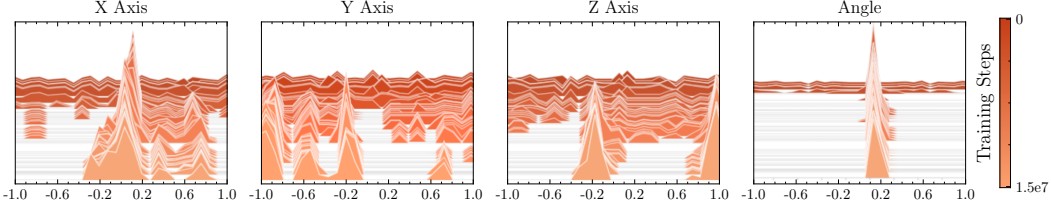

Figure 3: Evolution of how particles are distributed along four action dimensions during training of the DeepMimicWalk task with DPPO. The depicted four dimensions represent the target right hip joint position expressed in an axis-angle representation. Each action dimension is normalized between -1 and 1. Particles are initially distributed uniformly along a dimension (dark colors) and their locations adaptively change as the policy network is trained (light colors). The training steps are measured by the number of samples exploited during training.

for learning motor tasks, Table 1 further compares the performance of PFPN and DISCRETE across different discretization resolutions (number of particles and number of bins, respectively). As it can be seen, PFPN performs better than DISCRETE for any given resolution.

Intuitively, increasing the resolution, and hence the number of atomic actions, helps both DISCRETE and PFPN. However, the more atomic actions employed the harder the optimization problem will be for both methods due to the increase of policy gradient variance (see Appendix D for theoretical analysis). This can also be verified empirically by the performance decrease in Roboschool tasks when transitioning from 10 to 150 particles. In the more complex DeepMimic tasks, using too few atomic actions can easily skip over optimal actions. Although performance improves when the action resolution increases from 5 to 10, DISCRETE cannot provide any stable training results as denoted by the high variance among different trials. In contrast, PFPN performs significantly better across all resolutions and can reach best performance using only 35 particles. And as the number of particles increases beyond 35, PFPN has only a slight decrease in performance. This is because the training of DeepMimic tasks was run long enough to ensure that it could reach a relatively stable final performance. We refer to Appendix I for the sensitivity analysis of PFPN with respect to the number of particles, where employing more atomic actions beyond the optimal number results in slower convergence but similar final performance.

DeepMimic tasks rely on stable PD controllers and exploit motion capture data to design the reward function, which is more subtle than the Roboschool reward functions that primarily measure performance based on the torque cost and agent moving speed. Given a certain clip of motion, the valid movement of a joint may be restrained in some small ranges, while the action space covers the entire movement range of that joint. This makes position-based control problems more sensitive to the placement of atomic actions, compared to Roboschool torque-based control tasks in which the effective actions (torques) may be distributed in a relatively wide range over the action space. While uniform discretization could place many atomic actions blindly in bad regions, PFPN optimizes the placement of atomic actions, providing a more effective discretization scheme that reaches better performance with fewer atomic actions. As an example, Figure 3 shows how particles evolve during training for one of the humanoid's joints in the DeepMimicWalk task where PFPN reaches a cumulative imitation reward much higher than DISCRETE or Gaussian. We can see that the final active action spaces cover only some small parts of the entire action space. In Appendix H, we also show the particle evolution results for Roboschool's AntBulletEnv-v0 task. Here, the active action spaces are distributed more uniformly across the entire action space, which explains why DISCRETE is able to exploit its atomic actions more efficiently than in DeepMimic tasks.

## 5.3 CONTROL QUALITY

To highlight the control performance of the adaptive discretization scheme, Figure 4 compares the motion generated by PFPN and DISCRETE in the DeepMimicWalk task. As can be seen in the figure, PFPN generates a stable motion sequence with a nature human-like gait, while DISCRETE is stuck at a suboptimal solution and generates a walking motion sequence with an antalgic-like gait where the agent walks forward mainly through the use of its left leg. From the motion trajectories,

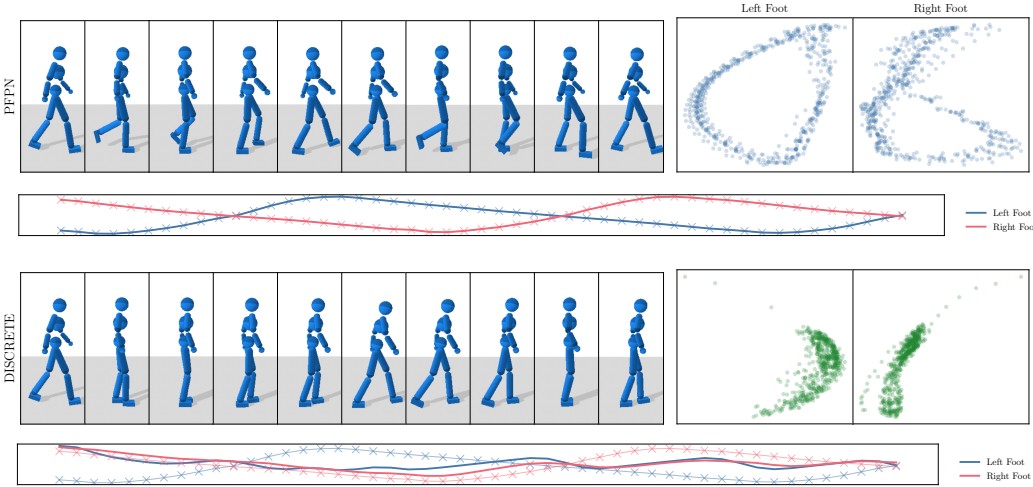

Figure 4: Comparison of motion generated by PFPN and DISCRETE in DeepMimicWalk task during one step cycle. Both PFPN and DISCRETE are trained with DPPO using the best resolution parameters from Table 1 (35 particles and 200 bins, respectively). PCA embedding of the trajectories of the agent's two feet are shown at the right and their time sequence expansions at the bottom of each method. Lines with "×" are the ground truth trajectories extracted from the motion capture data that the agent learns to imitate.

it can also be seen that PFPN results in a more stable gait with a clear cyclic pattern strictly following the motion capture data. We also assess the robustness of the learned policies to external perturbations. Table 2 reports the minimal force needed to push the humanoid agent down. All experiments were performed af-

| Task | Force Direction | PFPN | DISCRETE | Gaussian |
|---|---|---|---|---|
| DeepMimicWalk | Forward | **588** | 340 | 512 |
| | Sideway | **602** | 420 | 560 |
| DeepMimicPunch | Forward | **1156** | 480 | 720 |
| | Sideway | **896** | 576 | 748 |

Table 2: Minimal forwards and sideways push needed to make a DPPO DeepMimic agent fall down. Push force is in Newtons ($N$) and applied on the chest of the agent for 0.1s.

ter training using deterministic actions. It is evident that the PFPN agent can tolerate much higher forces than the Gaussian one. The fixed discretization in DISCRETE is less flexible, and as a result it cannot learn generalized features that will enable robust agent control. We also refer to Appendix H for additional results including motion trajectories obtained in the AntBulletEnv-v0 task, as well as the controls (torque profiles) of the corresponding ant agents.

## 6 CONCLUSION

We present a general framework for learning controls in high-dimensional continuous action spaces through particle-based adaptive discretization. Our approach uses a mixture distribution represented by a set of weighted particles to track the action policy using atomic actions during training. By introducing the reparameterization trick, the resulting particle-based policy can be adopted by both on-policy and off-policy policy gradient DRL algorithms. Our method does not change the underlying architecture or learning mechanism of the algorithms, and is applicable to common actor-critic policy gradient DRL algorithms. Overall, our particle filtering policy network combined with existing baselines leads to better performance and sampling efficiency as compared to corresponding implementations with Gaussian policies. As a way to discretize the action space, we show that our method is more friendly to policy gradient optimization by adaptive discretization, compared to uniform discretization, as it can optimize the placement of atomic actions and has the potential to meet the fine control requirement by exploiting fewer atomic actions. In addition, it leads to high quality motion and more robust agent control. While we currently track each action dimension independently, accounting for the synergy that exists between different joints has the potential to further improve performance and motion robustness, which opens an exciting avenue for future work.

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

## A  ALGORITHM

---

**Algorithm 1** Policy Gradient Method using Particle Filtering Policy Network

---

Initialize the neural network parameter $\theta$ and learning rate $\alpha$;
initialize particle parameters $\phi_i$ to uniformly distribute particles on the action dimension;
initialize the threshold $\epsilon$ to detect dead particles using a small number;
initialize the value of interval $n$ to perform resampling.
**loop**
  **for** each environment step **do**
    *// Record the weight while sampling.*
    $a_t \sim \pi_{\theta,\mathcal{P}}(\cdot|s_t)$
    $\mathcal{W}_i \leftarrow \mathcal{W}_i \cup \{w_i(s_t|\theta)\}$
  **end for**
  **for** each training step **do**
    *// Update parameters using SGD method.*
    $\phi_i \leftarrow \phi_i + \alpha \nabla J(\phi_i)$
    $\theta \leftarrow \theta + \alpha \nabla J(\theta)$
  **end for**
  **for** every $n$ environment steps **do**
    *// Detect dead particles and set up target ones.*
    **for** each particle $i$ **do**
      **if** $\max_{w_i \in \mathcal{W}_i} w_i < \epsilon$ **then**
        $\tau_i \sim P\left(\cdot|\mathbb{E}\left[w_k|w_k \in \mathcal{W}_k\right], k = 1, 2, \cdots\right)$
        $\mathcal{T} \leftarrow \mathcal{T} \cup \{\tau_i\}; \mathcal{D}_{\tau_i} \leftarrow \mathcal{D}_{\tau_i} \cup \{i\}$
      **end if**
    **end for**
    *// Resampling.*
    **for** each target particle $\tau \in \mathcal{T}$ **do**
      **for** each dead particle $i \in \mathcal{D}_\tau$ **do**
        *// Duplicate particles.*
        $\phi_i \leftarrow \phi_\tau$ with $\mu_i \leftarrow \mu_\tau + \varepsilon_i$
        *// Duplicate parameters of the last layer in the policy network.*
        $\boldsymbol{\omega}_i \leftarrow \boldsymbol{\omega}_\tau; b_i \leftarrow b_\tau - \log(|\mathcal{D}_\tau| + 1)$
      **end for**
      $b_\tau \leftarrow b_\tau - \log(|\mathcal{D}_\tau| + 1)$
      $\mathcal{D}_\tau \leftarrow \emptyset$
    **end for**
    $\mathcal{T} \leftarrow \emptyset; \mathcal{W}_i \leftarrow \emptyset$
  **end for**
**end loop**

---

## B  POLICY NETWORK LOGITS CORRECTION DURING RESAMPLING

**Theorem 1.** *Let $\mathcal{D}_\tau$ be a set of dead particles sharing the same target particle $\tau$. Let also the logits for the weight of each particle $k$ be generated by a fully-connected layer with parameters $\boldsymbol{\omega}_k$ for the weight and $b_k$ for the bias. The policy $\pi_\theta^\mathcal{P}(a_t|\mathbf{s}_t)$ is guaranteed to remain unchanged after resampling via duplicating $\phi_i \leftarrow \phi_\tau, \forall i \in D_\tau$, if the weight and bias used to generate the unnormalized logits of the target particle are shared with those of the dead one as follows:*

$$\boldsymbol{\omega}_i \leftarrow \boldsymbol{\omega}_\tau; \quad b_i, b_\tau \leftarrow b_\tau - \log\left(|\mathcal{D}_\tau| + 1\right). \tag{10}$$

*Proof.* The weight for the $i$-th particle is achieved by softmax operation, which is applied to the unnormalized logits $L_i$, i.e. the direct output of the policy network:

$$w_i(s_t) = \text{SOFTMAX}(L_i(s_t)) = \frac{e^{L_i(s_t)}}{\sum_k e^{L_k(s_t)}}. \tag{11}$$

Resampling via duplicating makes dead particles become identical to their target particle. Namely, particles in $\mathcal{D}_\tau \cup \{\tau\}$ will share the same weights as well as the same value of logits, say $L'_\tau$, after

resampling. To ensure the policy identical before and after sampling, the following equation must be satisfied

$$\sum_k e^{L_k(s_t)} = \sum_{\mathcal{D}_\tau \cup \{\tau\}} e^{L'_\tau(s_t)} + \sum_{k \notin \mathcal{D}_\tau \cup \{\tau\}} e^{L_k(s_t)} \tag{12}$$

where $L_k$ is the unnormalized logits for the $k$-th particle such that the weights for all particles who are not in $\mathcal{D}_\tau \cup \{\tau\}$ unchanged, while particles in $\mathcal{D}_\tau \cup \{\tau\}$ share the same weights.

A target particle will not be tagged as dead at all, i.e. $\tau \notin \mathcal{D}_k$ for any dead particle set $\mathcal{D}_k$, since a target particle is drawn according to the particles' weights and since dead particles are defined as the ones having too small or zero weight to be chosen. Hence, Equation 12 can be rewritten as

$$\sum_{i \in \mathcal{D}_\tau} e^{L_i(s_t)} + e^{L_\tau(s_t)} = (|\mathcal{D}_\tau| + 1)e^{L'_\tau(s_t)}, \tag{13}$$

Given that $e^{L_i(s_t)} \approx 0$ for any dead particle $i \in \mathcal{D}_\tau$ and that the number of particles is limited, it implies that

$$e^{L_\tau} \approx (|\mathcal{D}_\tau| + 1)e^{L'_\tau(s_t)}. \tag{14}$$

Taking the logarithm of both sides of the equation leads to that for all particles in $\mathcal{D}_\tau \cup \{\tau\}$, their new logits after resampling should satisfy

$$L'_\tau(s_t) \approx L_\tau(s_t) - \log(|\mathcal{D}_\tau| + 1). \tag{15}$$

Assuming the input of the full-connected layer who generates $L_i$ is $\mathbf{x}(s_t)$, i.e. $L_i(s_t) = \boldsymbol{\omega}_i \mathbf{x}(s_t) + b_i$, we have

$$\boldsymbol{\omega}'_i \mathbf{x}(s_t) + b'_i = \boldsymbol{\omega}_\tau \mathbf{x}(s_t) + b_\tau - \log(|\mathcal{D}_\tau| + 1). \tag{16}$$

Then, Theorem 1 can be reached.

If we perform random sampling not based on the weights during resampling (see Appendix I), it is possible to pick a dead particle as the target particle. In that case

$$L'_\tau(s_t) \approx L_\tau(s_t) - \log(|D_\tau| + (1 - \sum_k \delta(\tau, \mathcal{D}_k))), \tag{17}$$

where $L'_\tau(s_t)$ is the new logits shared by particles in $\mathcal{D}_\tau$ and $\delta(\tau, \mathcal{D}_k)$ is the Kronecker delta function

$$\delta(\tau, \mathcal{D}_k) = \begin{cases} 1 & \text{if } \tau \in \mathcal{D}_k \\ 0 & \text{otherwise} \end{cases} \tag{18}$$

that satisfies $\sum_k \delta(\tau, \mathcal{D}_k) \leq 1$. Then, for the particle $\tau$, its new logits can be defined as

$$L''_\tau(s_t) \approx (1 - \sum_k \delta(\tau, \mathcal{D}_k))L'_\tau(s_t) + \sum_k \delta(\tau, \mathcal{D}_k)L_\tau. \tag{19}$$

Consequently, the target particle $\tau$ may or may not share the same logits with those in $\mathcal{D}_\tau$, depending on if it is tagged as dead or not.

## C  POLICY REPRESENTATION WITH ACTION BOUNDS

In off-policy algorithms, like DDPG and SAC, an invertible squashing function, typically the hyperbolic tangent function, will be applied to enforce action bounds on samples drawn from Gaussian distributions, e.g. in SAC, the action is obtained by $\mathbf{a}_t(\varepsilon, \mathbf{s}_t) = \tanh u_{t,k}$ where $u_{t,k} \sim \mathcal{N}(\mu_\theta(\mathbf{s}_t), \sigma_\theta^2(\mathbf{s}_t))$, and $\mu_\theta(\mathbf{s}_t)$ and $\sigma_\theta^2(\mathbf{s}_t))$ are parameters generated by the policy network with parameter $\theta$.

Let $\mathbf{a}_t = \{\tanh u_{t,k}\}$ where $u_{t,k}$, drawn from the distribution represented by a particle with parameter $\phi_{t,k}$, is a random variable sampled to support the action on the $k$-th dimension. Then, the probability density function of PFPN represented by Equation 3 can be rewritten as

$$\pi_\theta^{\mathcal{P}}(\mathbf{a}_t|\mathbf{s}_t) = \prod_k \sum_i w_{i,k}(\mathbf{s}_t|\theta)p_{i,k}(u_{t,k}|\phi_{i,k})/(1 - \tanh^2 u_{t,k}), \tag{20}$$

and the log-probability function becomes

$$\log \pi_\theta^{\mathcal{P}}(\mathbf{a}_t|\mathbf{s}_t) = \sum_k \log \left[ \sum_i w_{i,k}(\mathbf{s}_t|\theta)p_{i,k}(u_{t,k}|\phi_{i,k}) - 2\left(\log 2 - u_{t,k} - \text{softplus}(-2u_{t,k})\right) \right].$$
$$\tag{21}$$

## D    CONCRETE DISTRIBUTION

Concrete distribution was introduced by Maddison et al. (2016). It is also called Gumbel-Softmax and proposed by Jang et al. (2016) concurrently. Here, we directly give the definition of concrete random variables $X \sim \text{CONCRETE}(\alpha, \lambda)$ by its density function using the notion from Maddison et al. (2016) as below:

$$p_{\alpha,\lambda}(x) = (n-1)!\lambda^{n-1} \prod_{k=1}^{n} \left( \frac{\alpha_k x_k^{-\lambda-1}}{\sum_{i=1}^{n} \alpha_i x_i^{-\lambda}} \right), \tag{22}$$

where $X \in \{x \in \mathbb{R}^n | x_k \in [0,1], \sum_{k=1}^{n} x_k = 1\}$, $\alpha = \{\alpha_1, \cdots, \alpha_n\} \in (0, +\infty)^n$ is the location parameter and $\lambda \in (0, +\infty)$ is the temperature parameter. A sample $X = \{X_1, \cdots, X_n\}$ can be drawn by

$$X_k = \frac{\exp((\log \alpha_k + G_k)/\lambda)}{\sum_{i=1}^{n} \exp((\log \alpha_i + G_i)/\lambda)}, \tag{23}$$

where $G_k \sim \text{GUMBEL}$ i.i.d., or more explicitly, $G_k = -\log(-\log U_k)$ with $U_k$ drawn from $\text{UNIFORM}(0,1)$.

From Equation 23, $X$ can be reparameterized using the parameter $\alpha$ and $\lambda$, and gives us a relaxed version of continuous approximation to the one-hot categorical distribution supported by the logits $\alpha$. As $\lambda$ is smaller, the approximation is more discrete and accurate. In all of our experiments, we pick $\lambda = 1$.

For convenience, in Section 3.4, we use CONCRETE with parameter $w \in \{w_1, \cdots, w_n | w_k \in [0,1], \sum_{k=1}^{n} w_k = 1\}$ as the probability weight to support a categorical distribution instead of the logits $\alpha$. We can get $\alpha$ in terms of $w$ by

$$\alpha_k = \log(w_k/(1-w_k)). \tag{24}$$

Since $X \sim \text{CONCRETE}(\alpha, \lambda)$ is a relaxed one-hot result, we use $\arg \max X$ to decide which particle to choose in the proposed reparameterization trick.

## E    VARIANCE OF POLICY GRADIENT IN PFPN CONFIGURATION

Since each action dimension is independent to others, without loss of generality, we here consider the action $a_t$ with only one dimension along which $n$ particles are distributed and the particle $i$ to represent a Gaussian distribution $\mathcal{N}(\mu_i, \sigma_i^2)$. In order to make it easy for analysis, we set up the following assumptions: the reward estimation is constant, i.e. $A_t \equiv A$; logits to support the weights of particles are initialized equally, i.e. $w_i(s_t|\theta) \equiv \frac{1}{n}$ for all particles $i$ and $\nabla_\theta w_1(s_t|\theta) = \cdots = \nabla_\theta w_n(s_t|\theta)$; particles are initialized to equally cover the whole action space, i.e. $\mu_i = \frac{i-n}{n}$, $\sigma_i^2 \approx \frac{1}{n^2}$ where $i = 1, \cdots, n$.

From Equation 5, the variance of the policy gradient under such assumptions is

$$\begin{aligned}
\mathbb{V}[\nabla_\theta J(\theta)|a_t] &= \int \frac{A_t \sum_i p_i(a_t|\mu_t, \sigma_t) \nabla_\theta w_i(s_t|\theta)}{\sum_i w_i(s_t|\theta) p_i(a_t|\mu_t, \sigma_t)} a_t^2 \mathrm{d}a_t \\
&\propto \sum_i \nabla_\theta w_i(s_t|\theta) \int a_t^2 p_i(a_t|\mu_t, \sigma_t) \mathrm{d}a_t \\
&\stackrel{\propto}{\sim} \sum_i (\mu_i^2 + \sigma_i^2) \nabla_\theta w_i(s_t|\theta) \\
&\propto \sum_i \frac{(i-n)^2 + 1}{n^2} \\
&= \frac{n}{3} + \frac{7}{6n} - \frac{1}{2} \\
&\sim 1 - \frac{3}{2n} + O(\frac{1}{n^2}).
\end{aligned} \tag{25}$$

Given $\mathbb{V}[\nabla_\theta J(\theta)|a_t] = 0$ when $n = 1$, from Equation 25, for any $n > 0$, the variance of policy gradient $\mathbb{V}[\nabla J(\theta)|a_t]$ will increase with $n$. Though the assumptions usually are hard to meet perfectly in practice, this still gives us an insight that employing a large number of particles may result in more challenge to optimization.

This conclusion is consistent with that in the case of uniform discretization (Tang & Agrawal, 2019) where the variance of policy gradient is shown to satisfy

$$\mathbb{V}[\nabla_\theta J(\theta)|a_t]_{\text{DISCRETE}} \sim 1 - \frac{1}{n}. \qquad (26)$$

That is to say, in either PFPN or uniform discretization scheme, we cannot simply improve the control performance of the police by employing more atomic actions, i.e. by increasing the number of particles or using more bins in the uniform discretization scheme, since the gradient variance increases as the discretization resolution increases. However, PFPN has a slower increase rate, which implies that it might support more atomic actions before performance drops due to the difficulty in optimization. Additionally, compared to the fixed, uniform discretization scheme, atomic actions represented by particles in PFPN are movable and their distribution can be optimized. This means that PFPN has the potential to provide better discretization scheme using fewer atomic actions and thus be more friendly to optimization using policy gradient.

## F    MULTI-MODAL POLICY

In this section, we show the multi-modal representation capacity of PFPN on special designed tasks.

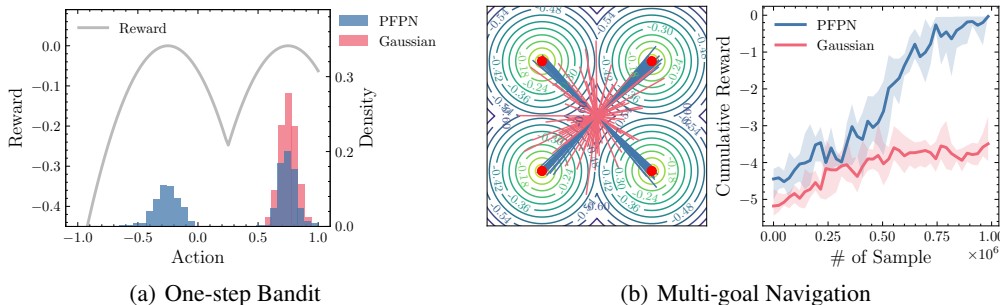

(a) One-step Bandit          (b) Multi-goal Navigation

Figure 5:    (a) One-step bandit task with asymmetric reward landscape. The reward landscape is defined as the gray line having two peaks asymmetrically at $-0.25$ and $0.75$. The probability densities of stochastic action samples drawn from PFPN (blue) and Gaussian policy (red) are counted after training with a fixed number of iterations. (b) Illustration of 2D multi-goal navigation. Left: one-step trajectories generated by PFPN and Gaussian policy via stochastic sampling after training. Red dots are the four goal points placed symmetrically and the contour line depicts the reward landscape by action costs proportional to the distance to the closest goal point. The Gaussian policy is initialized around the origin. Right: The learning curve of PFPN and Gaussian policy measured by cumulative rewards over five training trials with PPO algorithm.

**One-step Bandit.**    This is a simple task with one dimension action space $\mathcal{A} = [-1, 1]$. It has an asymmetric 2-peak reward landscape inversely proportional to the minimal distance to points $-0.25$ and $0.75$, as the gray line shown in Figure 5(a). The goal of this task is to find out the optimal points close to $-0.25$ and $0.75$. In Figure 5(a), we show the stochastic action sample distributions of PFPN and the naive Gaussian policy after training with the same number of iterations. It is clear that PFPN captures the bi-modal distribution of the reward landscape, while the Gaussian policy gives an unimodal distribution capturing only one of reward peaks.

**2D Multi-goal Navigation.**    In this task, the reward landscape is designed symmetrically with four goal points at $[\pm0.5, \pm0.5]$ and the agent has the task to reach any of the goal points. The cost (negative of the reward) function of the state $\mathcal{S} \in \mathbb{R}^2$ and action $\mathcal{A} \in \mathbb{R}^2$ is defined as the distance from the target position to the closet goal point. The naive Gaussian policy is initialized with mean value around the origin such that it is placed at the center point of the reward basin initially. Figure 5(b) shows the training performance curve and the one-step path trajectories given by PFPN and the Gaussian policy after training. While PFPN successfully learns diverse trajectories

reaching all four goals, the Gaussian policy fails to learn anything due to the symmetry of the reward landscape.

In the following, we limit our discussion to the expressivity of the basic action policy distributions generated directly by the policy network in actor-critic policy gradient methods, without the consideration of other methods to enhance the expressivity of action policies based on base ones, e.g. normalizing flows, or those to help better policy exploration by learning mechanisms, like entropy-based policies.

For simplicity, let us consider a 1-dimension Gaussian policy with log-probability of

$$\log \pi_\theta(a_t|\mathbf{s}_t) = -\frac{(a_t - \mu_\theta(\mathbf{s}_t))^2}{2\sigma_\theta^2(\mathbf{s}_t)} - \log \sigma_\theta(\mathbf{s}_t) - \log 2\pi. \tag{27}$$

where $a_t \sim \mathcal{N}(\mu_\theta(\mathbf{s}_t), \sigma_\theta^2(\mathbf{s}_t))$. In order to update the policy network with parameter $\theta$, we use Equation 2 with

$$\nabla_\theta \log \pi_\theta(a_t|\mathbf{s}_t) = \nabla_{\mu_\theta(\mathbf{s}_t)} \log \pi_\theta(a_t|\mathbf{s}_t) \nabla_\theta \mu_\theta(\mathbf{s}_t) + \nabla_{\sigma_\theta(\mathbf{s}_t)} \log \pi_\theta(a_t|\mathbf{s}_t) \nabla_\theta \sigma_\theta(\mathbf{s}_t) \tag{28}$$

where

$$\nabla_{\mu_\theta(\mathbf{s}_t)} \log \pi_\theta(a_t|\mathbf{s}_t) = \frac{(a_t - \mu_\theta(\mathbf{s}_t))}{\sigma_\theta^2(\mathbf{s}_t)}; \nabla_{\sigma_\theta(\mathbf{s}_t)} \log \pi_\theta(a_t|\mathbf{s}_t) = \frac{(a_t - \mu_\theta(\mathbf{s}_t))^2 - \sigma_\theta^2(\mathbf{s}_t)}{\sigma_\theta^3(\mathbf{s}_t)}. \tag{29}$$

The optimization of $\mu_\theta(\mathbf{s}_t)$ is along the direction of $A_t(a_t - \mu_\theta(\mathbf{s}_t))$. Therefore, the optimization process is analogous to sliding $\mu_\theta(\mathbf{s}_t)$, the location of policy distribution, towards or opposite to the sampled action $a_t$ regularized by the advantage estimation $A_t$. If $A_t$ is higher than the average $\mathbb{E}_t[A_t]$, the optimization will push $\mu_\theta(\mathbf{s}_t)$ towards to the location of $a_t$; if $A_t$ is smaller than the average, $\mu_\theta(\mathbf{s}_t)$ will be pushed away from the direction of $a_t - \mu_\theta(\mathbf{s}_t)$. Regarding the update of $\sigma_\theta(\mathbf{s}_t)$, $\sigma_\theta(\mathbf{s}_t)$ will increase and make the distribution cover $a_t$ if $A_t$ is higher than the average and the distance between $a_t$ and $\mu_\theta(\mathbf{s}_t)$ is out of the range of $\sigma_\theta(\mathbf{s}_t)$; otherwise, $\sigma_\theta(\mathbf{s}_t)$ will decrease and make the distribution shrink. Such an optimization strategy will encounter problems when, for example, facing a symmetric reward landscape defined by $A_t$, just like the 2D multi-goal navigation problem shown above. The Gaussian policy distribution in such a case would be optimized to move towards multiple directions simultaneously. This makes the policy distribution staying around its current location and unable to be optimized. Similarly, when facing an asymmetric, multimodal reward landscape as shown in the one-step bandit test case, the Gaussian policy would only be able to capture just one mode at best due to its unimodality.

Though in high-dimensional control problems, the reward landscape has a more complex shape, the multimodality is quite common in various, practical tasks. As an empirical proof, Figure 3 shows the evolution of the particle distributions during training, which captures the multimodality along different dimensions in DeepMimicWalk task. In such a task, Gaussian policy may face the problem of premature convergence and be stuck at suboptimal solutions because of its unimodality. Therefore, a multimodal policy distribution should be preferred for better capturing the reward landscape shape.

Fixed, uniform discretization (DISCRETE) schemes can also provide a multimodal policy represented by the categorical distribution. However, it is hard for DISCRETE to find an optimal solution due to the blindness of uniform discretization. Without loss of generality, we define the action space $\mathcal{A}$ as the range of $[-1, 1]$ and assume that the action space is discretized with $n$ bins. Under this setup, we will have atomic actions of $\{a_{t,1}, a_{t,2}, \cdots, a_{t,n}\}$ with a distance of $\frac{2}{n}$ between two consecutive atomic actions. Let $a_t^*(\mathbf{s}_t)$ be the optimal solution given the state $\mathbf{s}_t$. We can measure the error between the optimal solution $a_t^*(\mathbf{s}_t)$ and the best solution that DISCRETE could provide by

$$\min_k |a_t^*(\mathbf{s}_t) - a_{t,k}| \leq \frac{1}{n}. \tag{30}$$

This only gives us an upper bound of the minimal error as the half size of a bin. We can decrease the upper bound by exploiting more atomic actions. However, this cannot necessarily help decrease the lower bound. This analysis gives us an insight of the limitations that DISCRETE has.

Table 3 reports the lower bound of the minimum error that DISCRETE would achieve with varying number of bins. We can see that the best discretization scheme for DISCRETE is to use 4 bins.

| Number of Bins | 1 | 2 | 3 | 4 | 5 | 6 | 7 | 8 | 9 | 10 |
|---|---|---|---|---|---|---|---|---|---|---|
| Lower Bound Error | 0.25 | 0.25 | 0.08 | 0 | 0.05 | 0.08 | 0.036 | 0.125 | 0.03 | 0.05 |

Table 3: Lower bound of the minimum error that DISCRETE attains with different action resolutions in the one-step bandit task.

However, in practice, we are usually unable to quantitatively analyze the lower bound error with respect to the number of bins in complex DRL tasks. Increasing, though, the number of bins would reduce the possible maximum error. For example, it is better to use 9 bins compared to the case with 7 or 5 bins, or better to use 10 bins compared to that using 8 or 6 bins. However, the policy gradient variance problem caused by introducing more atomic actions (See Appendix E) would influence the optimization negatively. To solve this dilemma, PFPN can optimize the placement of atomic actions during training and thus has the potential to meet the fine control requirement with fewer atomic actions.

## G  HYPERPARAMETERS

| Parameter | Value |
|---|---|
| *Shared* | |
| optimizer | Adam (Kingma & Ba, 2014) |
| activation function | ReLU |
| resampling interval | 25 environment episodes |
| dead particle detection threshold ($\epsilon$) | 0.05/# of particles per action dimension |
| clip range (PPO/DPPO) | 0.2 |
| GAE discount factor (PPO/DPPO, A2C/A3C, $\lambda$) | 0.95 |
| truncation level (IMPALA, $\bar{c}, \bar{\rho}$) | 1.0 |
| reply buffer size (SAC, DDPG) | $1 \cdot 10^6$ |
| *Roboschool Environments* | |
| learning rate | $3 \cdot 10^{-4}$ |
| weight initializer | Orthogonal (Saxe et al., 2013) |
| number of neurons in hidden layers | [256, 256] |
| number of particle per action dimension | 35 (*Humanoid*), 10 (*others*) |
| discount factor ($\gamma$) | 0.99 |
| coefficient of policy entropy loss term (A2C/A3C, IMPALA) | 0.01 |
| *DeepMimic Environments* | |
| learning rate | $1 \cdot 10^{-4}$ |
| weight initializer | Truncated Normal with std. dev. of 0.05 |
| number of neurons in hidden layers | [1024, 512] |
| number of particle per action dimension | 35 |
| discount factor ($\gamma$) | 0.95 |
| coefficient of policy entropy loss term (A3C, IMPALA) | 0.00025 |

Table 4: Default Hyperparameters in Baseline PFPN Benchmarks

Table 4 lists the default hyperparameters used in all of our experiments. Regarding PPO and A2C, in all Roboschool tasks except for the HumanoidBulletEnv-v0 one, we use a single worker thread; for HumanoidBulletEnv-v0 and DeepMimic tasks, we exploit the advantage of distributed training and use DPPO (synchronous PPO) and A3C (asynchronous A2C) with multiple worker threads, while IMPALA is natively multi-thread.

# H ADDITIONAL RESULTS

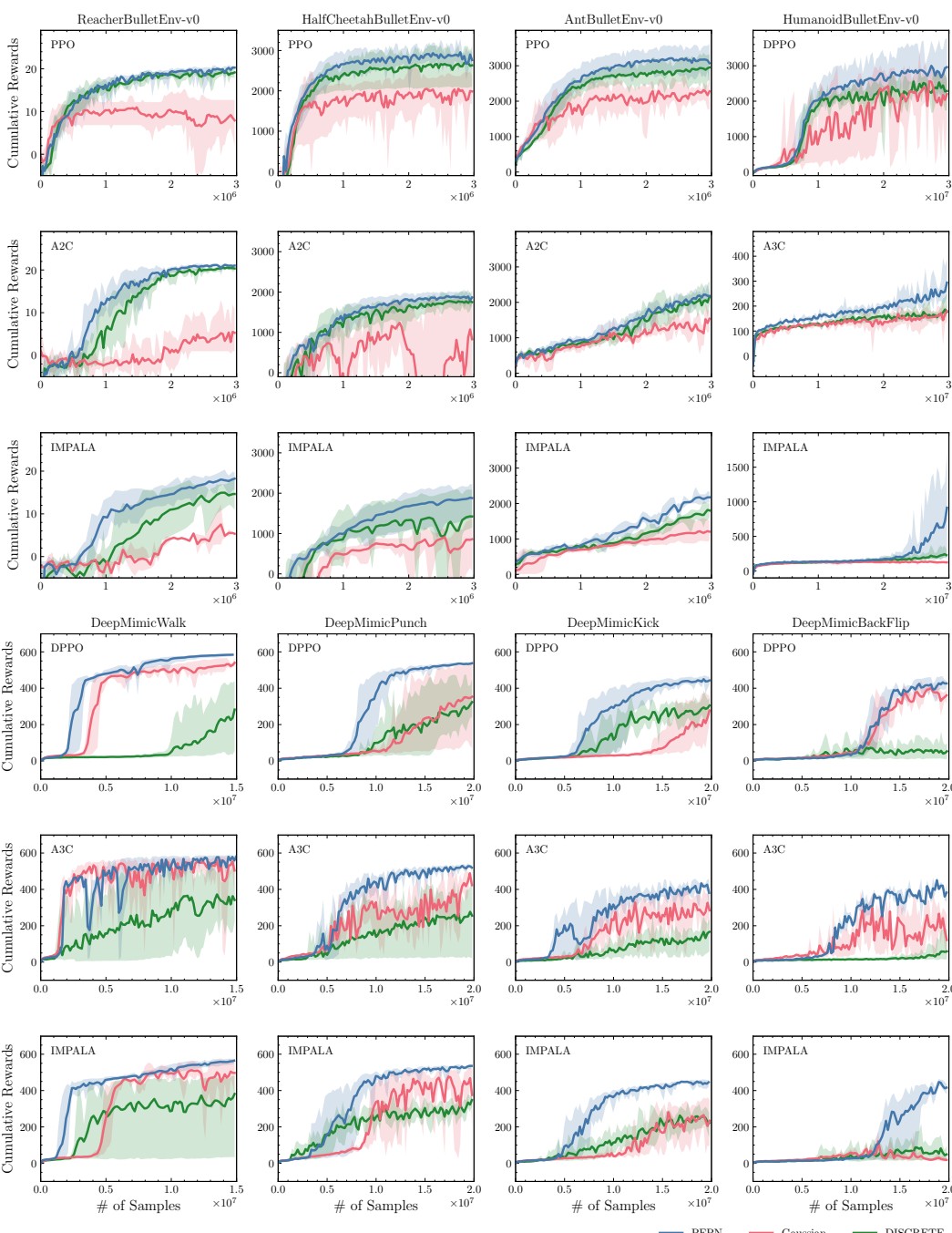

Figure 6: Training curves on continuous control tasks from the Roboschool and DeepMimic environments using on-policy policy gradient algorithms and IMPALA with v-trace correction.

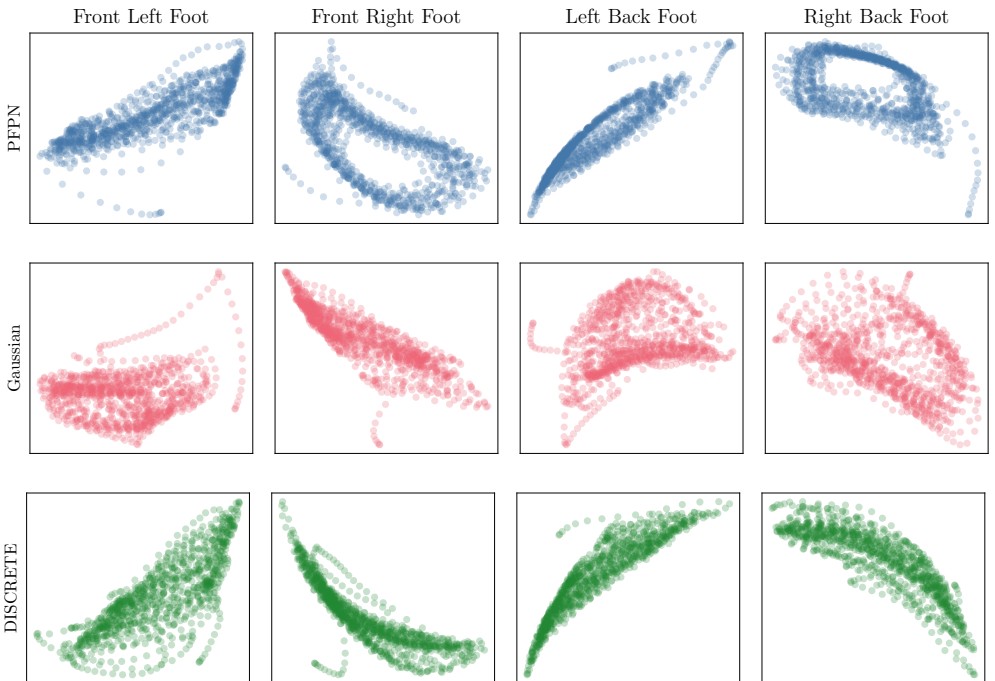

Figure 7: Motion trajectories of end effectors (four feet) of the ant agent in AntBulletEnv-v0 task with PFPN-PPO, Gaussian-based PPO and PPO with fixed, uniform discretization (DISCRETE). We apply PCA to visualize the 2D trajectories; the trajectories are measured by the relative positions of effectors with respect to the root link of the agent.

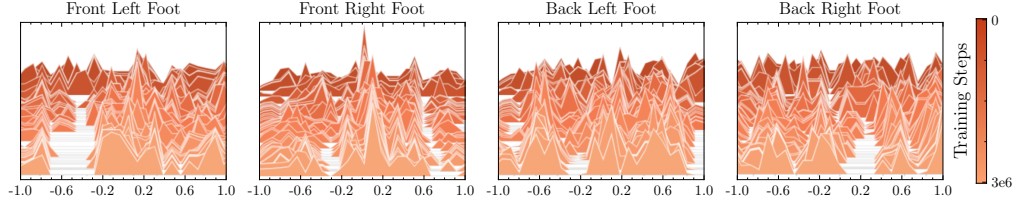

Figure 8: Evolution of particles distributed on action dimensions for the four feet joints of the agent in AntBulletEnv-v0 task during training using PPO.

### H.1 COMPARISON TO GAUSSIAN AND UNIFORM DISCRETIZATION POLICIES

Figure 6 compares baselines that employ Gaussian policies to their PFPN counterparts on a variety of Roboschool and DeepMimic tasks and the fixed, uniform discretization scheme (DISCRETE) with the same number of atomic actions. The results are obtained as discussed in Section 5.1, where ten evaluation trials run every 1,000 training steps using deterministic actions. PFPN outperforms Gaussian policies and fixed, uniform discretization scheme in those baselines and is more stable across different training trials.

To further highlight the value that particle-based discretization adds to the action exploration problem, we also compare the motion generated by a PFPN-PPO policy in the AntBulletEnv-v0 task to the ones obtained by vanilla PPO with Gaussian action policy and that using fixed, uniform discretization scheme. We project the one-episode motion trajectories of the four end effectors (feet) of the ant agent into two dimensions in Figure 7. As it can be seen, there is a significant difference in the motion taken by the three agents, with PFPN leading to higher performance as shown in Figure 7. Similarly to our analysis in Figure 4, Gaussian-based PPO and DISCRETE-PPO have more noise in the generated manifold, while PFPN-PPO performs robustly with a more clear cyclic

motion pattern. We also show the particle evolution of the ant agent during training in Figure 8, which reflects the optimization of particle distribution.

In Figure 9, we compare the torques generated by PFPN to those generated by Gaussian and DIS-CRETE. As it can been seen, PFPN generates significantly different torque profiles than the ones obtained by Gaussian both in terms of frequency and/or amplitude that result in higher asymptotic performance shown in Figure 6. PFPN has similar torque frequency and amplitude with DISCRETE, but also has significant difference in details, which leading to a better performance.

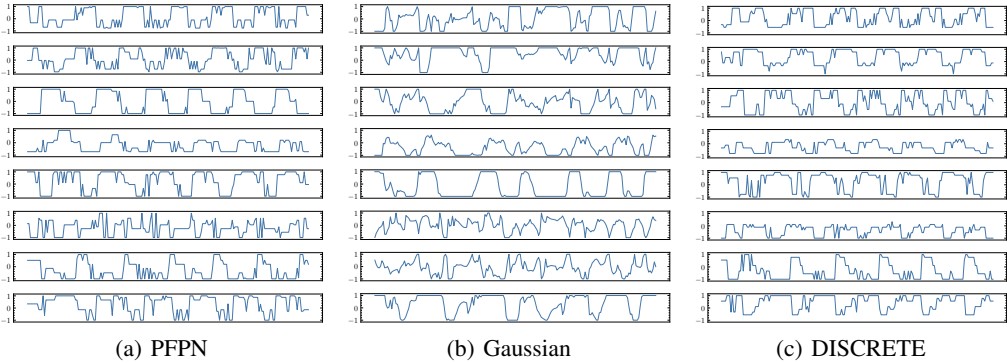

(a) PFPN  (b) Gaussian  (c) DISCRETE

Figure 9: Torque patterns generated by Gaussian-based PPO and PFPN-PPO on AntBulletEnv-v0. Each row denotes the corresponding policy per action dimension executed for 200 frames. Torque values are normalized to lie between -1 and 1.

## H.2    ADDITIONAL OFF-POLICY RESULTS

Besides the SAC benchmarks shown in Section 5.1, we test PFPN with DDPG, another popularly used off-policy algorithm, which runs policy gradient method only through state-action value. Differentiate the above experiments where each particle is assigned with a learnable variable and represents a Gaussian distribution, we follow the default configuration of DDPG and use a scalar noise during action sampling. DDPG provides relatively worse performance compared to SAC and cannot work well in HumanoidBulletEnv-v0 and DeepMimic tasks. As supplement, three environments, HalfCheetah-v2, Ant-v2 and Humanoid-v2, from OpenAI Gym benchmark suite are introduced in Figure 10. PFPN shows its advantage in most of those tasks.

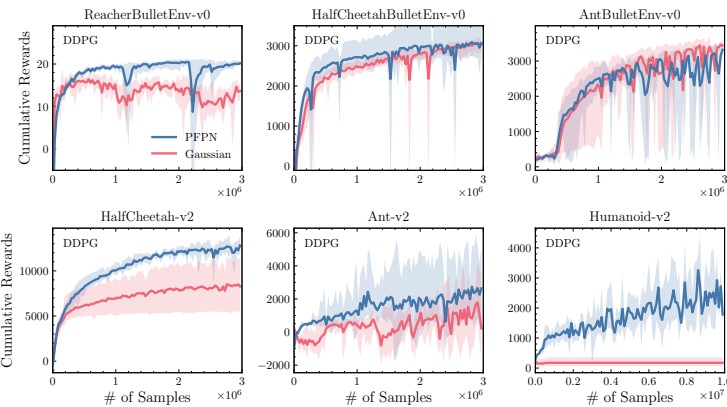

Figure 10: Learning curves on continuous control tasks using DDPG in Roboschool and OpenAI Gym environments.

| Environment | PFPN | DISCRETE | Gaussian | Gaussian-Big | GMM |
|---|---|---|---|---|---|
| AntBulletEnv-v0 | **3163 ± 323** (93,856) | 2863 ± 281 (93,776) | 2327 ± 199 (75,272) | 2398 ± 107 (94,352) | 2813 ± 380 (134,896) |
| HumanoidBulletEnv-v0 | **2840 ± 480** (121,172) | 2351 ± 343 (121,002) | 2462 ± 195 (81,681) | 2519 ± 219 (125,521) | 2026 ± 346 (208,382) |
| DeepMimicWalk | **584 ± 4** (1,375,192) | 245 ± 164 (1,373,932) | 540 ± 19 (746,020) | 517 ± 33 (1,380,498) | 203 ± 135 (2,666,692) |
| DeepMimicPunch | **537 ± 7** (1,375,192) | 317 ± 76 (1,373,932) | 359 ± 181 (746,020) | 309 ± 140 (1,380,498) | 210 ± 75 (2,666,692) |

Table 5: Performance comparison of PPO/DPPO using PFPN, uniform discretization (DISCRETE), Gaussian baselines (Gaussian), Gaussian baselines with a larger policy network (Gaussian-Big), and policies of fully state-dependent Gaussian mixture model (GMM). Reported numbers denote final performance averaged over 5 trials ± std after training is done. In each test case, the number of the policy network parameters is listed in parentheses below the reported performance.

### H.3 COMPARISON TO GAUSSIAN POLICIES WITH BIG NETWORK

The policy network in PFPN has similar size with the one in DISCRETE. Therefore, the performance advantage of PFPN over DISCRETE clearly comes from its more flexible discretization scheme. Compared to the Gaussian baselines, PFPN exploits a relatively larger policy network, though the number of hidden neurons is the same. To demonstrate that the performance gain of PFPN is not due to its large network size, we increase the number of hidden neurons in Gaussian baselines such that they have similar number of policy network parameters with PFPN. The performance comparison between PFPN and the scaled up version of Gaussian baselines (Gaussian-Big) is shown in Table 5. Training is done after a fixed number of samples are collected as shown in Figure 1. As can be seen from the table, increasing the number of hidden neurons does not lead to any significant improvements in Gaussian policies, with PFPN still reaching better final performance.

### H.4 COMPARISON TO FULLY STATE-DEPENDENT GAUSSIAN MIXTURE POLICIES

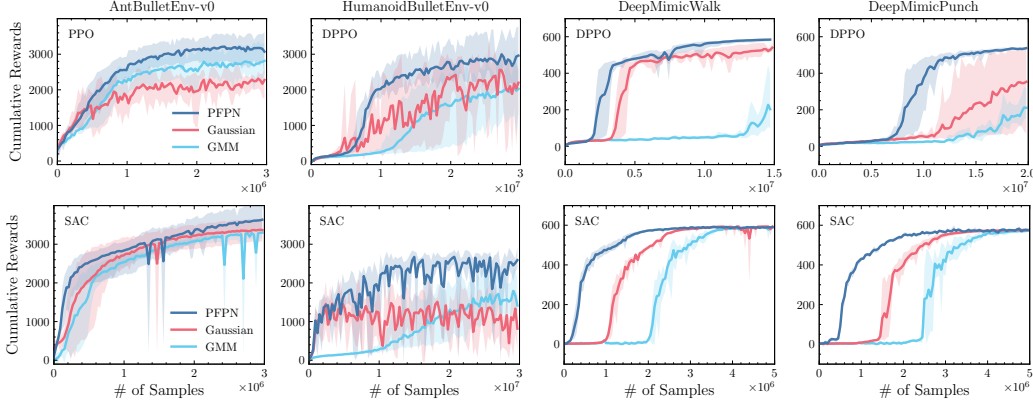

Figure 11: Comparison of PFPN to Gaussian baselines and fully state-dependent mixture of Gaussians (GMM). GMM in SAC uses the reparameterization trick described in Section 3.4 for state-action value based optimization.

PFPN in our experiments employs a mixture of Gaussians with state-dependent weights but state-independent components. The state-independent components can provide a global configuration of particle distributions that can be used as a discretization scheme. In theory, a fully state-dependent Gaussian mixture model (GMM) can also work as the action policy in DRL. However, in our experiments depicted in Figure 11, we found that GMM usually does not work quite well and even worse than Gaussians for complex continuous control problems. This is consistent with the results reported by Tang & Agrawal (2018a). As shown in Figure 11, GMM can only reach better performance than Gaussian in the AntBulletEnv-v0 task with PPO algorithm and in the HumanoidBulletEnv-v0 task using SAC, and is worse than PFPN in all tested benchmarks. GMM needs a much larger policy network as shown in Table 5. This may pose a challenge to optimization. Another issue of GMM

observed during our experiments is that GMM components are easy to collapse together and lose the advantage of multimodality. In the test cases of SAC, we use the reparameterization trick introduced in Section 3.4 to perform state-action value based optimization.

## H.5 TIME COMPLEXITY

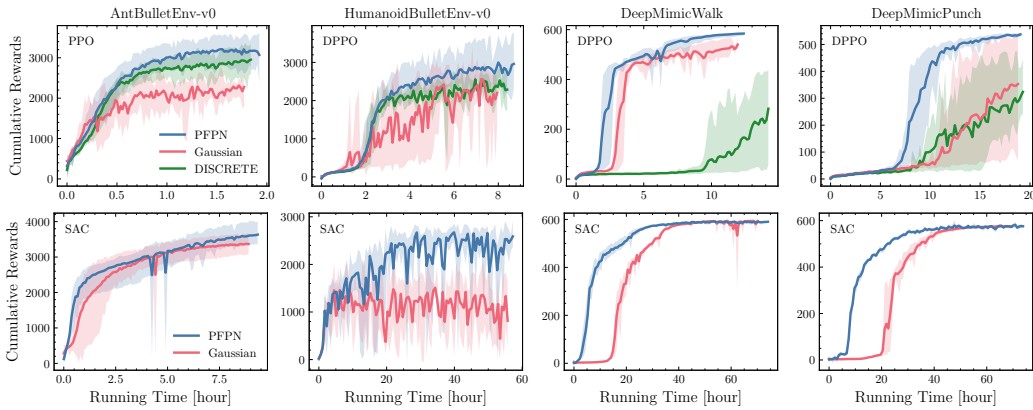

Figure 12: Cumulative rewards obtained during training as a function of the actual wall clock time. Training stops when a fixed number of samples is collected as reported in Figures 1 and 2.

Sampling from categorical distributions directly or using Gumbel tricks (Papandreou & Yuille, 2011) is typically more expensive than that from Gaussians. In addition, both PFPN and DISCRETE need more time than the Gaussian baselines in order to perform action sampling and complete a training iteration due to their larger network sizes. Compared to DISCRETE, PFPN has higher time complexity as it needs an extra process of sampling from the Gaussian distributions represented by the particles. While the resampling process can cost time as well, in practice, it is only performed every tens of thousand interactions with the environment during training. Despite its higher time complexity, though, PFPN is more sample efficient than Gaussian baselines and DISCRETE, needing less actual clock time to reach better performance. To highlight this, Figure 12 reports the training performance as a function of running time (wall clock time). All training was done on machines having the same configuration with Nvidia Tesla V100 GPUs and stopped when a fixed number of samples was collected for training.

PFPN needs more time to leverage the same number of samples as Gaussian baselines. However, the performance obtained at a fixed time can be significantly higher as shown in Figure 12. Especially in DeepMimic tasks, the sampling efficiency of PFPN is quite apparent. In DeepMimicWalk using DPPO, PFPN needs about 6 hours to reach a cumulative reward of 500, while Gaussian baselines needs around 8 hours to obtain a similar score. The gap between PFPN and Gaussian baselines is more evident in SAC cases. As our SAC implementation used a single thread, it took much more time to finish the training compared to DPPO that worked distributedly. While PFPN and Gaussian baselines using SAC have almost the same final performance, PFPN converged about ten hours before the Gaussian baseline. Simulation in DeepMimic tasks is quite expensive in term of wall clock time, mainly due to the complexity of running stable PD controllers (Tan et al., 2011) and computing the reward by comparing the agent motion to the motion capture data. Once the simulated humanoid agent falls down (one of the termination conditions in the training of DeepMimic tasks), the simulation environment needs to reset and start a new environment episode. This process can take lots of time and poses a challenge to the algorithm's ability to do better action space exploration at the initial stage of training. It is also the reason why DISCRETE needs more wall time than PFPN, even though its time complexity is lower.

## I SENSITIVITY ANALYSIS

**Number of Particles.** Since the particle configuration in PFPN is state-independent, it needs a sufficient number of particles to meet the fine control demand. Intuitively, employing more particles

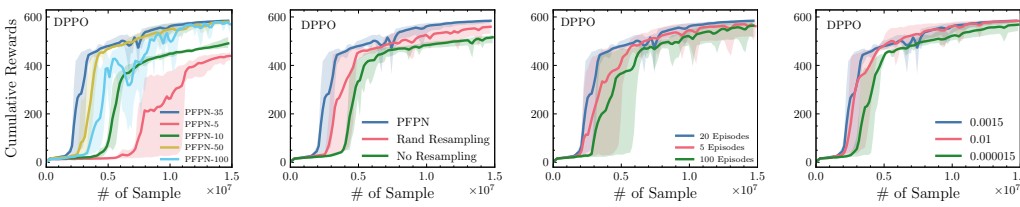

(a) Number of Particles  (b) Resampling Strategy  (c) Resampling Frequency  (d) Resampling Threshold

Figure 13: Sensitivity of PFPN to the number of particles and resampling strategies and hyper-parameters on DeepMimicWalk task using DPPO. (a) Comparison of PFPN using 35 particles per action dimension, which is the default parameters used in above benchmarks, to that using 5 (red), 10 (green), 50 (yellow) and 100 (azure) particles. (b)-(d) show the performance of PFPN with 35 particles on each action dimension but different resampling strategies or hyperparameters, where the blue line is the default hyperparameters used in our benchmark tests.

will increase the resolution of the action space, and thus increase the control capacity and make fine control more possible. However, in Appendix E, we prove that due to the variance of policy gradient increasing as the number of particles increases, the more particles employed, the harder the optimization would be. Therefore, it may negatively influence the performance to employ too many particles. This conclusion is consistent with the benchmark results of Roboschool tasks shown in Table 1 as the final performance decreases while the number of particles increases. The Deep-MimicWalk task with more than 35 particles, though reaching similar final performance, gets slower convergence as the number of particles increases. To clarify this, we show the learning curve of the DeepMimicWalk task with different number of particles in Figure 13(a), where PFPN-x represents the case that there are x particles per action dimension. As it can be seen, there is an obvious convergence delay as the number of particles per action dimension increases from 35 to 100. On the other hand, as we can see, a coarse discretization resolution employing too few particles, i.e., 5 or 10 particles per action dimension, will cause performance drops evidently, since the too few number of particles limits the control capacity.

**Resampling Strategies.** In Figure 13(b), We compare PFPN with default resampling strategy explained in Section 3.3 to a random resampling strategy (red) and that without resampling (green). The default resampling strategy is to draw targets for dead particles according to the weights of remaining alive ones. The random resampling strategy is to draw targets randomly from remaining alive particles. It can be seen that reactivating dead particles by resampling could help improve the training performance. Even the random resampling, though probably reactivating dead particles and place them on suboptimal locations, it still leads to better performance by keeping exploiting more effective particles, compared to the case without resampling. However, random resampling could lead to high variance and make the training process unstable by introducing too much uncertainty.

Figure 14 further compares PFPN's resampling strategy to no resampling for Roboschool and Deep-Mimic environments. As can be seen, in the more challenging DeepMimic tasks, our resampling strategy leads to an obvious improvement (note that 600 is the best reward that an algorithm can achieve). As compared to position-based DeepMimic control tasks, Roboschool agents directly learn torque-based controls where the effective atomic action space is uniformly distributed across the specified joint torque limits (see Section 5.2 for details). Taking also into account that PFPN requires only 10 atomic actions per dimension in the Roboschool tasks, particles are less likely to become degenerate since each of them will possibly become active at a certain observed state during training. While employing more particles would trigger the resampling process, this could lead to worse performance because of the increased difficulty in optimization (See Appendix E).

**Resampling Hyperparameters.** We also analyze the sensitivity of PFPN to the resampling hyperparameters: the resampling threshold $\epsilon$, which determines whether a particle is dead or not, and the resampling interval, which denotes how often the resampling is performed and is measured by environment episodes. Results in Figure 13(c) and 13(d) show that the resampling process itself is robust and not very sensitive to these two hyperparameters. However, an extremely small value of

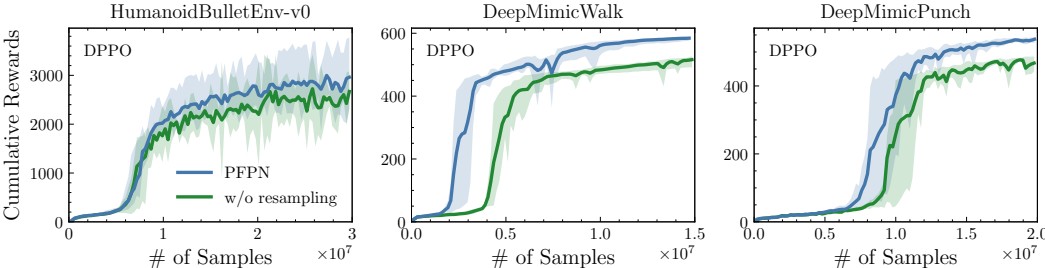

Figure 14: Learning curves of PPO/DPPO using PFPN with the default resampling process (blue) compared to that without resampling (green).

resampling threshold or large value of resampling interval still would hurt the performance by preventing resampling and push the learning curves towards to the case of PFPN without resampling. On the other hand, a too large resampling threshold will lead alive particles to being labeled as dead, increasing the variance of the training performance. Resampling with a too small interval will incur resampling before sufficient action exploration and thus cause the same problem with a too large value of resampling interval. In our tests, we choose 20 environment episodes as the default resampling interval, and a dynamic value of resampling threshold depending on the number of particles that each action dimension has, which is around 0.0015 in DeepMimic tasks with 35 particles per action dimension.

