# OpenReview forum: "Adaptive Discretization for Continuous Control using Particle Filtering Policy Network"
_ICLR.cc/2021/Conference — Reject_

### Official Review · AnonReviewer3 · 2020-10-28
**Adaptive Discretization for Continuous Control using Particle Filtering Policy Network**

**Rating:** 7
**Confidence:** 3

**Review:**

Post rebuttal update: The authors have addressed my concerns and the revised submission is much clearer, so I'm increasing my score to 7.


I must preface this with a caveat that I am not up to date with the latest results in continuous deep RL. That being said I think this is a nice and reasonably well-written paper, albeit with some weaknesses.

Major comments:

The biggest weakness of this paper appears to be that the results are only minor improvements (if they improve at all over the baseline). This is a major shortcoming because the proposed method is significantly more complicated than the baselines. If this additional complexity, and the additional hyper-parameters, does not bring major benefits then this begs the question as to whether or not it is worth it. On this point, what is the additional computational cost of the proposed method over the baselines?
If there is some other advantage to using this proposed method over the others then this should be explained more clearly up-front. I think a 'Contributions' section that listed the major takeaways of the paper would be useful.

Practically none of the mathematical quantities are defined correctly. If you have a mathematical object it should be defined as a member of a particular space, e.g., x in R^n, or s \in \S, or \pi \in \Delta_A. Otherwise it's very hard to reason about what these objects are. This is a major clarity problem.

Minor comments:

Why is eq 6 the product over the action dimensions 'k'? Isn't each dimension independent in this setup? The current formulation makes the policy appear to be a scalar and I was under the impression that it was a vector, although as previously mentioned this isn't clear.

The notation in eq 9 (mu and sigma on top of each other) is very strange, probably better to use \theta = (\mu, \sigma).

I don't see what you used for the At estimate. Is that explained anywhere?

I don't fully understand section 3.4, it's quite unclear as written. Why is the two-step sampling method 'non reparameterizable'? What does 'non reparameterizable' even mean? This needs a serious definition. Also the 'concrete' distribution is not defined here, it needs to be discussed more generally for completeness, possibly in the appendix is space is an issue.

Why is your method 'more friendly to policy gradient optimization' than simpler methods? Can this be quantified some way more concretely?

A few missing related works occurred to me (from the policy gradient / value learning literature):
https://arxiv.org/abs/1606.02647
https://arxiv.org/abs/1611.01626
http://papers.nips.cc/paper/6870-bridging-the-gap-between-value-and-policy-based-reinforcement-learning.pdf

Conclusion and next steps for the authors:

Overall I liked the paper, but cannot recommend it for publication in it's current form.

If the paper is improved to be clearer I will increase my score, in particular it needs better explanation of the concepts in some sections and every mathematical object should be defined rigorously. I dropped a point for clarity alone.

If the significance of the work, and the computational results can be better explained or improved I could increase my score again.

---

> ### Author Response · Authors · 2020-11-18
>
> - __“The biggest weakness of this paper appears to be that the results are only minor improvements (if they improve at all over the baseline).”__
> Please refer to the new Figure 1, Table 1, and Figures 6 & 10 (Appendix H) for a detailed comparison between PFPN and the baselines. The old Figure 2 only compared best rollout performance averaged over 5 trials rather than average performance which could be misleading. In all of our experiments, PFPN is as good as or significantly better than the baselines as measured by final performance and/or speed of convergence. In addition, it is more stable across different trials.
> Besides better performance and/or training stability, PFPN results in high-quality motion as shown in Figure 4 (also added comparisons to the ground truth motion captured data) and Figure 7, and more systematic exploration of the action space as shown in Figures 3 and 8. In addition, as shown in the new Table 2, PFPN policies are more robust to external perturbations.
> We clarify this in the updated Section 5 of the paper along with adding a "Contributions" paragraph in the introduction section.
>
>
> - __“additional computational cost of the proposed method over the baselines”__
> Despite its higher time complexity, PFPN can reach better performance in less wall clock time as compared to the other baselines. In the DeepMimicWalk task, for example,  SAC-PFPN needs about 10 fewer hours to converge than vanilla SAC. We refer to Appendix H.5 in our revised submission for the corresponding plots and further details.
>
>
> - __definition of mathematical quantities__
> We updated the text in our revised submission to clearly define the mathematical quantities of our proposed PFPN approach.
>
>
> - __“Why is eq 6 the product over the action dimensions 'k'? Isn't each dimension independent in this setup?”__
> Given a multi-dimensional action $\mathbf{a} = [a_1, a_2,\cdots, a_m]$, the distribution of $\mathbf{a}$ is the joint distributions of $a_1, a_2, \cdots, a_m$. Each dimension is independent, which means that $\mathbf{a}$ is a joint distribution for independent variables of $a_1, a_2, \cdots, a_m$. Therefore, the probability of the action vector $\mathbf{a}$ is:
> $p(\mathbf{a}) = p(a_1, a_2, ..., a_m) = p(a_1) p(a_2) ... p(a_m) = \prod_k p(a_k)$.
> This overcomes the curse of dimensionality, allowing the joint policy in PFPN to be tractable.
>
>
> - __“The notation in eq 9 (mu and sigma on top of each other) is very strange, probably better to use \theta = (\mu, \sigma).”__
> We updated the mathematical symbols in Equation 4 of the revised version (old Equation 9)  and in all other related equations. The definition is provided at the beginning of Section 3.1 to make the symbol representation consistent and clear.
>
>
> - __“what you used for the At estimate. Is that explained anywhere?”__
> Our approach only focuses on the action policy representation, namely the probability distribution used to represent $\pi_\theta(\mathbf{a}_t | \mathbf{s}_t)$ in Eq. 2. There is no special requirement for $A_t$. It can be chosen depending on the specific policy gradient algorithms. In our experiments, we used the generalized advantage estimator [1] (https://arxiv.org/pdf/1506.02438.pdf) in  PPO/DPPO, and V-trace based TD error [2] (https://arxiv.org/pdf/1802.01561.pdf) in IMPALA. We discuss this in Section 3.2 of our revised submission.
>
>
> - __“Why is the two-step sampling method 'non reparameterizable'? What does 'non reparameterizable' even mean?”__
> A probability distribution is “reparameterizable”  when given a sample drawn from the distribution, we can explicitly rewrite it as a function with respect to the parameters of that distribution. For example, the Gaussian distribution $\mathcal{N}(\mu, \sigma^2)$, is reparameterizable, since given a sampling result $a$, we can rewrite it in the form of $a = \mu + \epsilon \sigma$, where $\epsilon$ is a scalar.
> In the state-action value based optimization, say $Q(s, a)$, we need the action policy distribution to be reparameterizable so that we can write the drawn sample, a, as a result of differentiable function, i.e. $a = f_\theta(s)$ where $\theta$ is parameters that determine the distribution and $f_\theta(\cdot)$ is a differentiable function.  Then, the policy network with parameter $\theta$ can be updated by backpropagation through the gradient $\nabla_a Q(s, a) \nabla_\theta f_\theta(s)$.
> The two-step sampling method is non-reparameterizable, because the first step to draw samples from a categorical distribution is non-reparameterizable. The sample result drawn from a categorical distribution is an index number of a category. It cannot be directly rewritten as the result of a differentiable function with respect to the logits or probability weights that support the categorical distribution.
> To clarify this, we add a brief explanation in the background section of our new submission.

---

> > ### Author Response · Authors · 2020-11-18
> >
> > - __“the 'concrete' distribution is not defined”__
> > The concrete distribution was proposed by Maddison et al. [3]. (https://arxiv.org/pdf/1611.00712.pdf). It is also called Gumbel-Softmax and proposed by Jang et al. [4] (https://arxiv.org/pdf/1611.01144.pdf) concurrently.
> > Given a categorical distribution supported by a set of weights $\mathbf{w} =$ {$w_1, w_2, \cdots, w_k$}, where $\sum_i w_i = 1$, $\mathbf{x} \sim Concrete(\mathbf{w}, \lambda)$ will give us a vector $\mathbf{x} =$ {$x_1, x_2, \cdots, x_k$} where $x_1, x_2, \cdots, x_k$ are all scalar numbers satisfying $\sum_i x_i = 1$ and differentiable to the parameters $\mathbf{w}$; by doing multiple sampling, the distribution of all the x will be close to the distribution of the corresponding categorical distribution supported by $\mathbf{w}$. The parameter $\lambda$ controls the degree of the approximation.
> > We added a brief introduction about this in Appendix D of our revised submission.
> >
> >
> > - __“Why is your method 'more friendly to policy gradient optimization' than simpler methods? Can this be quantified some way more concretely?”__
> > Our PFPN approach is more friendly to policy gradient optimization than uniform discretization schemes (DISCRETE), because it can optimize the placement of atomic actions, and thus meet the fine control demand of continuous control problems by exploiting fewer actions. In contrast, DISCRETE can only increase the number of atomic actions to make fine control possible, and this could make the optimization problem harder due to the increase in policy gradient variance. (The problem of policy gradient variance is discussed in Appendix D.)
> > As an example, in DeepMimic tasks, PFPN with even five atomic actions per dimension can outperform all DISCRETE test cases with atomic actions from 5 to 400 per dimension. This means that PFPN can effectively exploit the five atomic actions through optimizing their placement. See Table 1 and Figure 3 for related results.
> > We further elaborate on this in the new Section 5.2. We have also updated the conclusion section accordingly.
> >
> >
> > - __“A few missing related works”__
> > We have added the references to these works in our revised submission.
> >
> >
> > [1] Schulman, J., Moritz, P., Levine, S., Jordan, M. and Abbeel, P., 2015. High-dimensional continuous control using generalized advantage estimation. arXiv preprint arXiv:1506.02438.
> > [2] Espeholt, L., Soyer, H., Munos, R., Simonyan, K., Mnih, V., Ward, T., Doron, Y., Firoiu, V., Harley, T., Dunning, I. and Legg, S., 2018. Impala: Scalable distributed deep-rl with importance weighted actor-learner architectures. arXiv preprint arXiv:1802.01561.
> > [3] Maddison, C.J., Mnih, A. and Teh, Y.W., 2016. The concrete distribution: A continuous relaxation of discrete random variables. arXiv preprint arXiv:1611.00712.
> > [4] Jang, E., Gu, S. and Poole, B., 2016. Categorical reparameterization with gumbel-softmax. arXiv preprint arXiv:1611.01144.

---

### Official Review · AnonReviewer2 · 2020-10-28
**Review 2**

**Rating:** 5
**Confidence:** 3

**Review:**

Summary
----------

This paper presents an approach to multimodal policies based on Gaussian mixtures. The policy is parameterized as a set of Gaussian distributions (with state-invariant mean and variance) weighted by state-dependent mixture weights, which are the output of a (softmaxed) network. The weighting network and the means and covariances of the Gaussians are updated with standard RL losses. The authors propose a resampling scheme for mixture elements that consistently have low weight, in the style of resampling in particle filters. The authors evaluate the method with several RL algorithms including PPO and SAC, and on a variety of environments.

Comments
----------

Overall, this paper is pointing in an interesting direction. Expressive action distributions are often an afterthought in RL (although they have seen increasing attention with SAC), and this work is an interesting step in that direction. However, there are several limitations that, if addressed, would strengthen the paper substantially.

Overall, the evaluation is quite thorough. However, the authors do not include a comparison to a standard mixture of gaussians policy, in which the Gaussians mixture elements are also state-dependent. The authors state that the mixture of Gaussians presented "cannot support any discretization scheme". What is meant by this? The PFPN approach does not actually "discretize" the action space, it simple uses a mixture of Gaussians to parametrize the policy to yield a multimodal action distribution. Thus, the fully state-dependent mixture of Gaussians seems like the most relevant comparison for the PFPN approach, and should be able to rely on the sample Gumbel-softmax trick for differentiable sampling.

The most interesting contribution of the paper is the resampling scheme. However, there is minimal evaluation of the benefit of this scheme. While section H of the appendix is a good start, I would expect a much more thorough investigation of the resampling, as it is probably the most substantial novel contribution of this work. Indeed, the current results for the resampling are somewhat worrying, as they show only a minor performance drop for removing the resampling entirely. An evaluation of the value of resampling (and the hyperparameters associated with resampling) on a variety of environments would strength the paper.

Finally, the paper should better address the value of expressive multimodal policies beyond just performance increases in standard environments. This is addressed briefly in section E of the appendix, but a broader discussion of the benefits of the approach would help make clear the benefits associated with the PFPN approach.

Post-Rebuttal
----------

I thank the authors for their response. The additional baseline comparisons do strengthen the paper, and I have increased my score from 4 to 5. I agree with the authors that the mixture of Gaussians policy is substantially weaker than their method, and is a useful baseline experiment to have.

However, the added experiments with random sampling are somewhat worrying---the performance improvement of the proposed re-sampling scheme is quiet minor over random resampling. In the future, the authors may want to investigate the random resampling for the systems in figure 14.

---

> ### Author Response · Authors · 2020-11-18
>
> - __“comparison to a standard mixture of Gaussians policy, in which the Gaussians mixture elements are also state-dependent”__
> We ran comparisons between PFPN and a fully state-dependent mixture of Gaussians (GMM) with our reparameterization trick using PPO and SAC as our baselines. Please see the corresponding results in the new Appendix H.4. Overall, the performance of GMM is not good in most test cases, especially in high-dimensional control tasks. This conclusion is consistent with the results reported by Tang & Agrawal [1] (https://arxiv.org/pdf/1809.10326.pdf), as well as by the fact that the final version of SAC replaced the GMM action policy with a squashed Gaussian. GMM needs a policy network with many more parameters and this could make optimization become hard. In addition, Gaussian components in GMM can easily collapse together due to their state-dependent nature.
>
> - __“the mixture of Gaussians presented  ‘cannot support any discretization scheme’”__
> Indeed, the statement above is confusing. To clarify, the fully state-dependent mixture of Gaussians cannot provide a discretization scheme that is consistent globally for all given states, since we cannot check all possible states to decide the possible placement of all the atomic actions. In our approach, particles are distributed state-independently and deterministically after training, and thus they can be exploited to support a discretization scheme.
> To clarify this, we modified accordingly the related text in the related work section of our submission.
>
> - __“a much more thorough investigation of the resampling strategy... on a variety of environments”__
> In Appendix I, we compare the performance of the default resampling strategy of PFPN to that without resampling. To show the advantage of resampling, we also introduced the case of “random” resampling. The default resampling strategy draws targets for dead particles according to the weights of remaining alive ones. The random resampling strategy uses the same resampling method but draws targets randomly from all the alive ones. Our goal is to show that the resampling approach, even when using the random strategy, helps improve the performance as compared to no resampling; and, overall, the default resampling strategy is more robust and efficient than the random one. We note that in Roboschool tasks where PFPN requires only 10 particles, no resampling is needed as all particles remain active. However, in the more challenging DeepMimic tasks, our resampling strategy leads to an obvious improvement compared to the case without resampling (note that 600 is the best reward that an algorithm can achieve). In general, as the number of particles increases, the resampling strategy is needed to ensure that all particles contribute to tracking the action policy distribution. We updated the text in Appendix I to clarify this along with adding the corresponding comparison figures.
>
>
> - __“better address the value of expressive multimodal policies beyond just performance increases in standard environments”__
> Besides better performance and/or training stability, PFPN results in higher quality motion as shown in Figure 4 (also added comparison to mocap data) and Figure 7, and more systematic exploration of the action space as shown in Figures 3 and 8. Related discussion is provided in the revised Section 5.2. In addition, as shown in the new Table 2, PFPN policies are more robust to external perturbations.  We highlight this in Section 5.3 of our revised submission, along with including an overall "Contributions" paragraph in the introduction section to summarize the benefits of the proposed PFPN approach. We also expand our discussion in Appendix F to provide a more thorough analysis of the issues that the Gaussian policy could face due to its unimodality, as well as the issues revolving around uniform discretization.
>
> [1] Tang, Y. and Agrawal, S., 2018. Boosting trust region policy optimization by normalizing flows policy. arXiv preprint arXiv:1809.10326.

---

### Official Review · AnonReviewer4 · 2020-10-29
**GMM Policy implemented by Particle Filtering**

**Rating:** 5
**Confidence:** 4

**Review:**

1. Optimal policy should be unique, and please make corrections for motivation. Flexible policy representation is helpful when the agent is not confident about the environment (exploration). The energy-based policy is helpful for efficient exploration, as it can better model the uncertainty of an agent.

2. Particle filtering is relatively computationally intensive. Could you please provide a comparison of the running time?

3. The main contribution of this paper is a GMM policy using particle filtering. This direction is not that exciting. The initial version of SAC does use the GMM, but the final version uses isotropic Gaussian. Particle-based variational inference with RL has also been explored by Stein Variational Policy Gradient and Policy Optimization as WGFs. It is good to see improvement empirically, but I am not sure the improvement source. An ablation study on this should be useful.

4. It is interesting to see the differentiable reparameterization trick. I cannot verify its correction. What does 'arg max x' mean?  If one prefers the flexible representation, the energy-based policy as in Soft Q-Learning should be a good choice. The dead particle issue will not happen as a repulsive force added in SVGD gradient step.

---

> ### Author Response · Authors · 2020-11-18
>
> 1. __Optimal policy should be unique, and please make corrections for motivation. Flexible policy representation is helpful when the agent is not confident about the environment (exploration). The energy-based policy is helpful for efficient exploration, as it can better model the uncertainty of an agent.__
> We updated the related text in our revised submission and state our motivation more clearly, as our work focuses on action policies defined by a multimodal distribution, which is more expressive than the unimodal Gaussian, without changing the underlying model architecture or learning mechanisms of policy gradient algorithms.  To that end we propose a particle-based action policy with a reparameterization trick that makes our approach applicable to both on-policy and off-policy policy gradient algorithms.
>
>
> 2. __Particle filtering is relatively computationally intensive. Could you please provide a comparison of the running time?__
> Despite its higher time complexity, PFPN can reach better performance in less wall clock time as compared to the other baselines. In the DeepMimicWalk task, for example,  SAC-PFPN needs about 10 less hours to converge than vanilla SAC. We refer to Appendix H.5 in our revised submission for the corresponding plots and further details.
>
>
> 3. __The initial version of SAC does use the GMM, but the final version uses isotropic Gaussian. Particle-based variational inference with RL has also been explored by Stein Variational Policy Gradient and Policy Optimization as WGFs. It is good to see improvement empirically, but I am not sure the improvement source. An ablation study on this should be useful.__
> Our approach focus on the representation of action policy, i.e., choosing a probability distribution to represent $\pi(a_t|s_t; \theta)$ in Eq.2. It does not take any variational inference technique and is compatible with SVPG, WGF or soft Q-learning, as those methods are based on a general definition of the action policy distribution. For example, our PFPN can be incorporated into Eqs. 2~4 in the SVPG paper [1] (https://arxiv.org/pdf/1704.02399.pdf) to generate $\pi(a_t|s_t; \theta)$; or as a replacement to $a=f^\phi(\xi;s_t)$ in the soft Q-learning paper [2] (https://arxiv.org/pdf/1702.08165.pdf).
> In the official implementation of many baselines, including soft Q-learning and SAC , Gaussian-based policies are the default choice dealing with continuous action space. We proposed to use a mixture of Gaussians as a replacement to the Gaussian policy and introduce a resampling method to address the problem of Gaussian component degeneracy. Meanwhile, to solve the problem of non-reparameterization of the mixture of Gaussians, we proposed a reparameterization trick to make our approach work for both on-policy and off-policy policy gradient methods. Therefore, we only compared our approach to Gaussian baselines, and, as our approach provides an adaptive action space discretization scheme, in case of on-policy methods to baselines that use a fixed, uniform discretization scheme.
> The mixture of Gaussians distribution that we exploit in the paper has state-independent components mixed by state-dependent weights. This is different to that presented in the first version of SAC, where the Gaussian mixture model is fully state-dependent. We do an ablation study of the resampling approach and sensitivity analysis in Appendix I. As requested by other reviewers, we compared our approach to the fully state-dependent Gaussian mixture models (GMM) in the new Appendix H.4. Our approach outperforms the GMM policies which in turn are outperformed by Gaussian policies.

---

> > ### Author Response · Authors · 2020-11-18
> >
> > 4. __What does 'arg max x' mean? If one prefers the flexible representation, the energy-based policy as in Soft Q-Learning should be a good choice. The dead particle issue will not happen as a repulsive force added in SVGD gradient step.__
> > The concrete distribution was proposed by Maddison et al. [3] (https://arxiv.org/pdf/1611.00712.pdf). It is also called Gumbel-Softmax and proposed by Jang et al. [4] (https://arxiv.org/pdf/1611.01144.pdf) concurrently.
> > Given a categorical distribution supported by a set of weights $w =$ {$w_1,\cdots, w_k$}, where $\sum_i w_i = 1$, $x \sim Concrete(w, \lambda)$ will give us a vector $x =$ {$x_1, \cdots, x_k$} where $x_1, \cdots, x_k$ are all scalar numbers satisfying $\sum_i x_i = 1$ and differentiable to the parameters w; by using multiple samples, the distribution of all the x will be close to the distribution of the corresponding categorical distribution supported by w.  The parameter $\lambda$ controls the degree of the approximation. We added a brief introduction about this in Appendix D of our revised submission.
> > Our approach focuses on a different direction than the SVGD method used in the SVPG algorithm. We focus only on the probability representation of $\pi(a_t|s_t;\theta)$. A particle in our approach only represents an atomic action on an action dimension, having a state-dependent importance weight. The placement of particles provides a scheme to discretize the action space, and our goal is to optimize the placement of particles adaptively during training. Given that $x \sim Concrete(w, \lambda)$ is a relaxed one-hot vector, we need an $\arg\max$ operation to decide which particle to choose in the proposed reparameterization trick.
> > A repulsive force can help prevent particles from collapsing together but cannot avoid dead particles in our approach. If, for example, a particle is initialized at a location where the reward is low, its weight will decrease during training; and, once the weight decreases to near zero, the particle is considered inactive because there is almost no choice to draw that particle when sampling and that particle cannot contribute towards tracking the policy. In Section 3.3, we proposed a method to reactivate the dead particles by resampling.
> >
> >
> >
> > [1] Liu, Y., Ramachandran, P., Liu, Q. and Peng, J., 2017. Stein variational policy gradient. arXiv preprint arXiv:1704.02399.
> > [2] Haarnoja, T., Tang, H., Abbeel, P. and Levine, S., 2017. Reinforcement learning with deep energy-based policies. arXiv preprint arXiv:1702.08165.
> > [3] Maddison, C.J., Mnih, A. and Teh, Y.W., 2016. The concrete distribution: A continuous relaxation of discrete random variables. arXiv preprint arXiv:1611.00712.
> > [4] Jang, E., Gu, S. and Poole, B., 2016. Categorical reparameterization with gumbel-softmax. arXiv preprint arXiv:1611.01144.

---

### Official Review · AnonReviewer1 · 2020-10-29
**The paper proposes an interesting method but should be more didactic on why it is not impacted by the issues that are affecting the other approaches.**

**Rating:** 4
**Confidence:** 3

**Review:**

In the paper "Adaptive Discretization for Continuous Control using Particle Filtering Policy Network", the authors introduce a new way to discretise the action space of agent in RL settings by using a Particule Filtering approach. The main idea is that the learned policy will output the weight of each particle to define which one should be used, while the position of the particle changes during the learning process. Particles that are not moved (because they have a weight that is systematically too low) are removed and resampled from other particles.

The paper is well structured and clearly illustrated.
However, I am unable to understand from the text why the proposed approach is not subject to the curse of dimensionality. In particular, if for every action dimension 10 to 35 weights should be defined by the policy, this creates a significant increase in the size of the search space (35^N). In particular, I would appreciate seeing a discussion about the size (in terms of the number of parameters) of the policies.

It is also quite surprising to see that the considered task in figure 2 can mostly be solved with very few particles (for both of the compared approaches). Therefore, I don't understand why in the last experiment of the paper (figure 3), the compared approaches use 35 vs. 200 particles. The complexity of the search space is certainly impacted by the difference and the observed results might just be the result of this. I am also wondering how the analysis can be extended to the other experiments. It seems that the performance difference on the other tasks is significantly more subtle and it will be important to discuss this.

Overall, the paper proposes an interesting method but should be more didactic on why it is not impacted by the issues that are affecting the other approaches.

---

> ### Author Response · Authors · 2020-11-18
>
> - __“why the proposed approach is not subject to the curse of dimensionality”__
> Both PFPN and DISCRETE represent the joint distribution over discrete actions as factorized across dimensions. Consequently, if we have N actions dimensions and M atomic actions per dimension, the joint policy is defined as $\pi(a|s) = \prod_{k=1}^N\pi_{\theta_k}(a_k|s)$ (see Eq. 3 in the revised submission). In PFPN, $\pi_{\theta_k}(a_k|s)$ is a sum of M weighted particles for a given dimension $k$, while in DISCRETE it denotes a categorical distribution over $M$ actions for a given dimension $k$. Hence, in PFPN, for a $N$-dimension action space with $M=35$ particles on each dimension, the policy network has $35N$ output neurons instead of $35^N$. A softmax operation is applied to every set of 35 neurons to compute the corresponding particle weights, and finally the policy network generates an $N$-dimensional vector as the sampled action. If the number of particles on each dimension increases, only the number of parameters of the last fully-connected layer has to change. Hence, in PFPN, the joint policy is tractable. We clarify this in the updated version of the paper (Section 3.1).
>
>
> - __“a discussion about the size (in terms of the number of parameters) of the policies”__
> The policy network in PFPN has almost the same size as the one used in categorical policy distributions (DISCRETE). The only difference is that PFPN has a state-independent configuration of particles that results in more robust training and improved performance as shown in Figures 1 & 6 and Table 1. The particles bring only a minor increase in the total number of parameters.
> Compared to Gaussian baselines, the policy network of PFPN has more parameters. However, the performance advantage of PFPN is not due to the large scale of its policy networks. In Appendix H.3 of the revised text, we report the number of policy network parameters of each baseline, and compare PFPN and Gaussian baselines with a big policy network where the number of parameters is of the same magnitude with PFPN.  Increasing the number of parameters in the policy network of Gaussian baselines did not result in any significant performance improvements, with PFPN still outperforming the Gaussian baselines. In the new Appendix H.5, we also compare the runtime performance of PFPN, Gaussian, and DISCRETE highlighting the sample  efficiency of our proposed approach.
>
>
> - __“the considered task in figure 2 can mostly be solved with very few particles”...“why in the last experiment of the paper (figure 3), the compared approaches use 35 vs. 200 particles”__
> We apologize for not clearly communicating this in our original submission. The old Figure 2 shows only the __best__ rollout performance averaged over 5 trials. In the revised version of the paper, we replaced it with the new Table 1 that focuses on final performance __averaged__ over multiple training trials as a function of action resolution, along with the new Figure 1 that reports performance for a fixed resolution. These results along with Figure 6 in Appendix H make clear that PFPN outperforms DISCRETE. Importantly, for the more challenging DeepMimic tasks, DISCRETE is significantly worse than PFPN and rather unstable as demonstrated by the high variance obtained by different methods.
> In DeepMimicWalk, DISCRETE needs 200 bins to exhibit a stable performance which is still worse than our approach both quantitatively and qualitatively as shown in the motion trajectories of Figure 4 in the new submission (old Figure 3). We clarify all this in the new version of the paper along with providing comparisons to the ground truth motion captured data.

---

> > ### Author Response · Authors · 2020-11-18
> >
> > - __“how the analysis can be extended to the other experiments. It seems that the performance difference on the other tasks is significantly more subtle and it will be important to discuss this.”__
> > While in AntBulletEnv-v0 tasks, PFPN and DISCRETE exhibit similar behavior (as measured by asymptotic performance, training speed, and variance across trials), this is not the case for the more challenging tasks, like all the three DeepMimic ones. This applies to all on-policy methods tested as can be seen from the learning curves of Figure 6 in Appendix H. The reason that performance differences are more apparent in DeepMimic tasks compared to Roboschool ones is due to the different action spaces that the two frameworks employ.
> > DeepMimic uses position-based controllers, where the action space is defined by the entire movement range of each agent's joint. However, the valid movement of a joint is typically restrained in some small range as shown in Figure 3 (the original Figure 4).  This makes position-based control problems more sensitive to the placement of atomic actions, compared to the Roboschool suite that focuses on torque-based control tasks. Here, the actions (torques) are distributed in a relatively wide range over the action space as shown in Figure 8, and thus uniform discretization can perform relatively well. We have revised Section 5.2 to better communicate all this.

---

### Author Response · Authors · 2020-11-18

Thanks for all your comments! We revised the paper based on your feedback. All changes and new results are highlighted in blue. Please let us know if there are more questions. In the meantime, we will briefly reply to some individual comments.

---

### Decision · Program_Chairs · 2021-01-07
**Final Decision**

**Decision:**

Reject

**Comment:**

This paper is rejected.

The authors contributions are:
* Propose PFPN as an expressive action policy for continuous action spaces.
* Introduce a reparameterization trick for training PFPN with off-policy methods.
* Experiments claiming PFPN outperforms unimodal Gaussian policies and a uniform discretization scheme and that it is more sample efficient and stable across different training trials.

I and the reviewers appreciate the additions by the authors. The GMM baseline is an important addition addressing concerns from several reviewers. However, I agree with R2's comment that "most interesting contribution of the paper is the resampling scheme. However, there is minimal evaluation of the benefit of this scheme [...] However, the added experiments with random sampling are somewhat worrying---the performance improvement of the proposed re-sampling scheme is quiet minor over random resampling. In the future, the authors may want to investigate the random resampling for the systems in figure 14." Without resampling, the proposed method is a location/scale state-independent GMM policy. It is interesting that this outperforms the fully state-dependent GMM, and the authors could investigate that further. To justify the additional complexity of the resampling step, the authors need to perform further investigation and move that to the main text.

In addition, the evaluated environments omit standard OpenAI gym environments (which the authors do have access to as evidenced by their experiments w/ DDPG on them in the Appendix). This makes evaluating baseline method performance challenging. Furthermore, the authors cite Tang & Agrawal (2018) which introduces a normalizing flow policy that outperforms GMM. It would be natural to compare to that baseline. Finally, Figurnov et al. (2018) among others shows how reparameterization gradients can be computed through GMMs. The authors should explain why this is not applicable.